# Convolution Goes Higher-Order: A Biologically Inspired Mechanism Empowers Image Classification

## Abstract

We propose a novel approach to image classification inspired by complex nonlinear biological visual processing, whereby classical convolutional neural networks (CNNs) are equipped with learnable higher-order convolutions. Our model incorporates a Volterra-like expansion of the convolution operator, capturing multiplicative interactions akin to those observed in early and advanced stages of biological visual processing. We evaluated this approach on synthetic datasets by measuring sensitivity to testing higher-order correlations and performance in standard benchmarks (MNIST, FashionMNIST, CIFAR10, CIFAR100 and Imagenette). Our architecture outperforms traditional CNN baselines, and achieves optimal performance with expansions up to $3^{rd}/4^{th}$ order, aligning remarkably well with the distribution of pixel intensities in natural images. Through systematic perturbation analysis, we validate this alignment by isolating the contributions of specific image statistics to model performance, demonstrating how different orders of convolution process distinct aspects of visual information. Furthermore, Representational Similarity Analysis reveals distinct geometries across network layers, indicating qualitatively different modes of visual information processing. Our work bridges neuroscience and deep learning, offering a path towards more effective, biologically inspired computer vision models. It provides insights into visual information processing and lays the groundwork for neural networks that better capture complex visual patterns, particularly in resource-constrained scenarios.

## 1 Introduction

*"The correlation over space translates into a correlation over order."* (Koenderink & van Doorn, 2018)

Convolutional Neural Networks (CNNs) have become the cornerstone of modern computer vision tasks, demonstrating remarkable performance across a wide range of applications (Krizhevsky et al., 2012), (Simonyan & Zisserman, 2014), (He et al., 2016). The success of CNNs is largely attributed to their ability to capture local patterns and hierarchical features in images through their layered structure and weight sharing mechanism (LeCun et al., 2015), (Zeiler & Fergus, 2014).

However, natural images contain complex correlations and higher-order statistics that extend beyond simple linear relationships (Field, 1993), (Olshausen & Field, 1996), (Schwartz & Simoncelli, 2001). These include texture patterns, edge interactions, and intricate geometric structures that are prevalent in real-world visual scenes. Importantly, these correlations exhibit a hierarchical structure, with higher-order correlations becoming increasingly sparse (Koenderink & van Doorn, 2018). Traditional CNNs, particularly those with limited depth, often struggle to effectively exploit these higher-order correlations (Gatys et al., 2016), (Ustyuzhaninov et al., 2022), (Ding et al., 2020). The limitation of standard CNNs in capturing higher-order correlations stems from their reliance on linear convolutions followed by pointwise nonlinearities (Yu & Salzmann, 2017). Although deeper networks can approximate complex functions through the composition of many layers, this approach may be computationally expensive and require large amounts of training data.

To address this limitation, we propose a learnable extension of the classical linear convolution operator to incorporate higher-order terms, akin to a Volterra expansion (Volterra, 1887), (Volterra, 1930). This approach allows for the direct modeling of multiplicative interactions between neighboring pixels, enabling the network to capture complex local structures more effectively, even in shallow architectures.

We design models by incorporating higher-order convolutions, and compare their performance against standard CNN baselines. We evaluated our model on a range of datasets, beginning with synthetic images typically used to assess sensitivity to higher-order correlations. Subsequently, we extended our experiments to compare our model against CNN baselines on widely-used datasets including MNIST, FashionMNIST, CIFAR10, CIFAR100 and Imagenette. Our findings reveal that these higher-order convolutions improve image classification performance, offering a promising direction for advancing deep learning models in computer vision tasks.

Interestingly, our results show optimal performance at the 3$^{rd}$ or 4$^{th}$ order, aligning remarkably with the distribution of pixel intensities in natural images as analyzed by Koenderink & van Doorn (2018). This alignment suggests that our approach effectively captures the fundamental structure of visual information in the natural world. Through systematic perturbation analysis, we isolate the contributions of specific image statistics to model performance, providing insight into how different orders of convolution process distinct aspects of visual information. Furthermore, we employ Representational Similarity Analysis (RSA) (Kriegeskorte et al., 2008) to investigate the internal representations learned by our model compared with standard CNNs, revealing distinct geometries that indicate different strategies for encoding and processing visual information.

Our approach addresses the limitations of simple pointwise nonlinearities in capturing local image structure, especially in networks of limited depth or width. In particular, we show that incorporating such computations at the earliest stages of image processing, even within the first layer of a neural network, yields significant benefits. This finding suggests that models benefit from embedding these computations via direct engagement with the *superficial structure* of images (Koenderink, 1984).

## 1.1 BIOLOGICAL INSPIRATION AND RELATED WORK

Our work draws inspiration from the sophisticated mechanisms of biological visual systems, where visual information processing extends beyond simple pointwise nonlinearities and first-order convolutions (Fitzgerald & Clark, 2015; Clark et al., 2014; Lettvin et al., 1959; Peterhans & Von der Heydt, 1993; Levitt et al., 1994). These non-pointwise nonlinearities, which consider spatial relationships between multiple input points, enable more effective extraction and integration of spatial information (Krieger et al., 1997; Zetzsche & Barth, 1990; Barth et al., 1993; Koenderink & Richards, 1988).

This complex processing begins in the retina, where certain cells exhibit responses to visual stimuli that cannot be explained by simple pointwise nonlinearities (Lettvin et al., 1959). In the visual cortex, specialized neurons respond to complex visual features like corners or line endings (Hubel & Wiesel, 1965), performing advanced computations that integrate information across their receptive fields. Such multidimensional, non-pointwise operators are fundamental to visual processing, as emphasized by Zetzsche et al. (1993) and Koenderink & Richards (1988). Similar mechanisms appear across species - fruit flies employ non-pointwise computations for motion detection as early as the second dendritic layer (Fitzgerald & Clark, 2015), suggesting these computations are fundamental for efficient visual processing (Gilbert, 2007). Recent research has further demonstrated that single dendrites can solve complex problems like XOR (Xu et al., 2012; Ran et al., 2020), a capability that pointwise nonlinearities struggle to model (Minsky & Papert, 1969).

In computational research, the concept of nonlinear receptive fields has been explored through various approaches. Berkes & Wiskott (2006) analyzed inhomogeneous quadratic forms as receptive fields, while Zoumpourlis et al. (2017) extended this to CNN-based learning. For video classification, Roheda et al. (2024) introduced Volterra Neural Networks using a low-rank approach. Related developments in fine-grained image classification include bilinear CNN models (Lin et al., 2015) and compact bilinear pooling (Gao et al., 2016), with applications extending to computational neuroscience (Ahrens et al., 2008).

Our approach differs fundamentally from previous methods by embedding higher-order operations within the convolutional operator itself, rather than applying them after feature extraction. This de-

sign choice enables the network to learn and apply complex nonlinear transformations throughout its entire depth, creating a more flexible architecture that better constitutes an *algorithmic implementation* of biological visual processing. By integrating these operations directly into the network's core building blocks, we enable more efficient feature learning while maintaining architectural simplicity.

### 1.2 BEYOND POINTWISE NONLINEARITIES

Traditional convolutional neural networks typically employ pointwise nonlinearities with weighted sums of inputs as arguments. Expanding these nonlinearities as a polynomial series reveals that different orders of the expansion share the same weights, effectively coupling the terms across orders and leading to inter- and extra-order dependencies:

$$\sigma\left(\sum_{i=0}^{2} w_i x_i\right) \approx \alpha_0 + \alpha_1(w_1 x_1 + w_2 x_2) + \alpha_2(w_1 x_1 + w_2 x_2)^2 + \ldots$$

$$= \alpha_0 + \alpha_1 w_1 x_1 + \alpha_1 w_2 x_2 + \alpha_2 w_1^2 x_1^2 + \alpha_2 w_2^2 x_2^2 + 2\alpha_2 w_1 w_2 x_1 x_2 + \ldots \tag{1}$$

This *tied-weight* issue becomes particularly problematic when the network is not *deep enough* or *wide enough* to compensate, as the precise bounds for *enough* depth or width depend on the universal approximation theorem and are often challenging to determine exactly (Bahri et al., 2020). Consequently, the standard pointwise nonlinearity with linear summation may fail to capture complex relationships in the data, especially in shallower or narrower networks.

To address this limitation, we propose a novel approach that can be viewed as an implementation of non-pointwise nonlinearities. Our method, which will be described in detail in **Section** 2, can be seen as a *learnable* generalized 2-D detector model (Zetzsche & Barth, 1990) representing a receptive field that responds to multiplicative interactions between inputs, similar to how some biological neurons process visual information (see **Subsection** 1.1). This approach offers a solution for capturing complex correlations at the level of a single mechanism, allowing for more flexible and powerful representations even in compact network architectures.

## 2 MODEL ARCHITECTURE: HIGHER-ORDER CONVOLUTION

To perform more complex computations while preserving the benefits of locality and weight sharing that are inherent in standard convolutions, we propose the concept of higher-order convolution. This approach extends the traditional convolution operator to include higher-order terms, allowing for more sophisticated feature extraction, and resulting in a stackable layer for empowering CNNs.

In order to explain our higher-order convolutional layer, we start by considering an input image patch **P** with $n$ elements, reshaped as a vector **x**. Standard convolution can be expressed in terms of linear filtering, relating input **x** and output **y** as follows: $y(\mathbf{x}) = b + \sum_{i=1}^{n} w_1^i x_i$ where $w_1^i, i = 1, ..., n$ represents the weights of the convolution kernel.

We expand this function to include quadratic, cubic, and higher-order terms. The general form of this expansion is:

$$y(\mathbf{x}) = b + \sum_{i=1}^{n} w_1^i x_i + \sum_{j=1}^{n} \sum_{i=1}^{n} w_2^{ij} x_i x_j + \sum_{k=1}^{n} \sum_{j=1}^{n} \sum_{i=1}^{n} w_3^{ijk} x_i x_j x_k + ... \tag{2}$$

Extending this to the entire image, we can express the higher-order convolution operation as:

$$Y_{l,m} = b + \sum_{i,j} w_1^{ij} X_{l+i,m+j}$$

$$+ \sum_{i,j} \sum_{k,h} w_2^{ijkh} X_{l+i,m+j} X_{l+k,m+h}$$

$$+ \sum_{i,j} \sum_{k,h} \sum_{p,q} w_3^{ijkhpq} X_{l+i,m+j} X_{l+k,m+h} X_{l+p,m+q}$$

$$+ \ldots \tag{3}$$

Here, $Y_{l,m}$ represents the output at position $(l, m)$ in the feature map, $X$ is the input image, and $w_1$, $w_2$, $w_3$ are the learnable weights for the first, second, and third-order terms respectively. The sums over $i$, $j$, $k$, $h$, $p$, $q$ run over the dimensions of the convolution kernels for each order.

By symmetry considerations, the total number of parameters in this expansion grows as: $n_V = \frac{(n+p)!}{n!p!}$ where $p$ represents the order of the term and $n$ the size of the kernel. To address potential issues with the relative magnitudes of higher-order terms, we implement a normalization strategy. For kernels of order greater than the 1$^{\text{st}}$, we apply a scaling factor $s$ calculated as $s = \frac{1}{\sqrt{n_V}}$. The scaling factor helps to balance the contribution of higher-order terms relative to the first-order (linear) terms, promoting stability and preventing higher-order terms from dominating network behavior during training.

**Figure** 1 **A** illustrates this concept for binary image, showing how the classical convolution is extended to include higher-order terms for a single patch. In practice, this operation is applied across the entire image (as shown in the equation above), resulting in a feature map for each order of the expansion, which are then summed together before a standard pointwise nonlinearity as ReLU (see for example, **Figure** 3 **B**).

At this point, we have outlined the construction of a higher-order convolutional layer. This layer can functionally replace a standard convolutional layer in neural network architectures and is compatible with training through backpropagation.

## 2.1 A QUANTITATIVE PERSPECTIVE ON THE TIED-WEIGHT ISSUE

To demonstrate how a Higher-order CNN (HoCNN) addresses the tied-weight issue introduced in **Section** 1.2, we conducted a systematic analysis of weight independence in the network's representations. We generated a fixed input image ($32 \times 32$ pixels) containing binary textures exhibiting all possible 1-, 2-, 3-, and 4-point correlations for 2×2 patches, corresponding to fixing the $x_i$ terms in Equation 1. We then randomly initialized the same model architecture 10,000 times and analyzed the activations after the convolutional block (Conv/HoConv + BatchNorm + ReLU + Max Pooling) for two architectures: a CNN with 10 kernels of size 2×2 and a HoCNN with 2 kernels of size 2×2 with higher-order expansions.

To quantify the degree of weight independence, we employed Principal Component Analysis of these activations, measuring the number of principal components (PCs) needed to explain 95% of the variance. Our hypothesis was that tied weights would reduce the number of principal components, while greater weight independence would require more components to capture the same variance. The results confirmed our expectations (see **Figure** 1 **B**): the CNN required only **0.9%** of PCs (87 out of 9610) to explain 95% of the variance, while the HoCNN required **5.3%** of PCs (102 out of 1922) with second-order expansion and **8.3%** (159 out of 1922) with third-order expansion. While the CNN has more total components due to its higher channel count (5×), additional analyses with matched channel counts yielded similar results.

These findings quantitatively demonstrate that higher-order convolutions introduce more independent weights, leading to a richer representation space with reduced weight coupling. The increasing number of components needed for variance explanation directly shows how our approach mitigates the tied-weight issue inherent in standard CNNs. Additional analyses with various nonlinearities showed consistent results, further supporting these conclusions (see **Appendix** A.5).

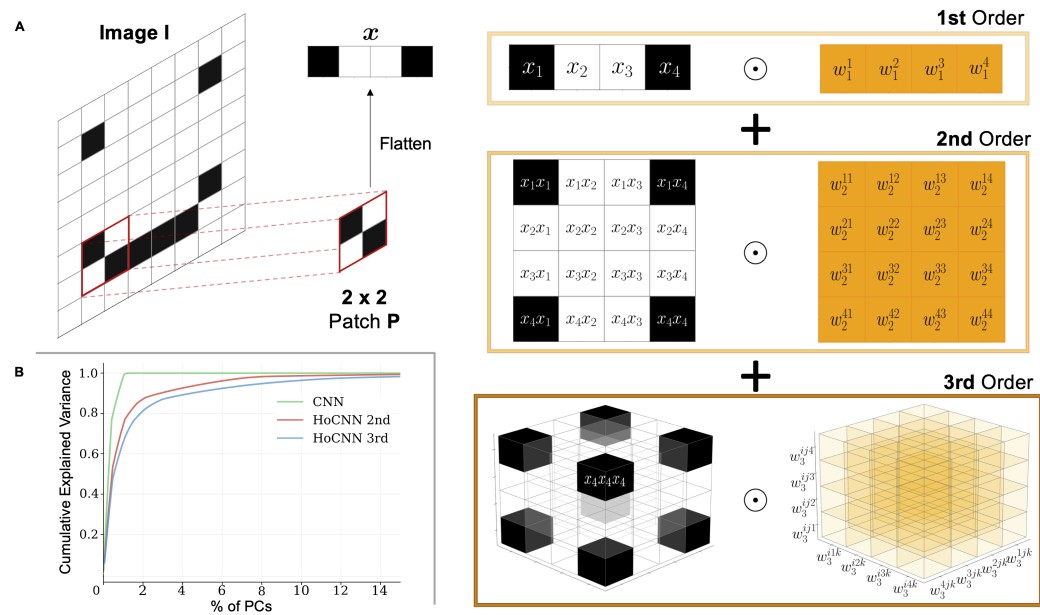

Figure 1: **Extending classical convolution** (A) A toy example of how higher-order convolution is implemented. (Left) We consider a patch located at the bottom-left of the input image, and by flattening it to a vector. (Right) Classical convolution would correspond to the 1$^{st}$ order term, where a set of weights (the kernel) are learnt and then translated onto other patches (weight-sharing). We extend classical convolution by expanding the first term: for the 2$^{nd}$ order, we compute the outer product of the original vector **x** with itself (thus obtaining a matrix) and the corresponding set of weights; similarly for the 3$^{rd}$ order, we iterate the same operation one more time and learn the associated set of weights. Feature maps of the three convolutions are then summed up before a standard pointwise nonlinearity as, for example, a ReLU function. (B) Cumulative explained variance of principal components for standard CNN model vs Higher-order CNN models with 2nd and 3rd expansion quantitatively confirms the tied-weight issue for classical models.

## 3 EXPERIMENTS & RESULTS

We evaluated the performance of our model on multiple datasets, including a synthetic one consisting of textures with higher order correlations and four widely-used benchmarks: MNIST, Fashion-MNIST, CIFAR-10, CIFAR-100 and Imagenette. This section details our experimental setup and findings.

### 3.1 SYNTHETIC DATASET: STRUCTURED VISUAL TEXTURES

To rigorously test the sensitivity of our model to higher-order correlations, we generated a synthetic dataset of structured visual textures based on the work of Victor & Conte (2012) (see **Figure** 2). These textures allow precise control over the type and intensity of contained correlations.

To generate the textures, we utilized a custom software library (Piasini et al., 2021) implementing the method developed by Victor & Conte (2012). This process involves sampling from a distribution of binary textures with specific multipoint correlation probabilities while maximizing entropy. The correlation intensity is parameterized by parity counts of white or black pixels within 1-, 2-, 3-, or 4-pixel tiles (termed *gliders*, **Figure** 2)). Notably, two-point and three-point gliders can produce multiple distinct multipoint correlations based on their spatial configurations. Two-point correlations can arise from horizontal, vertical, or oblique gliders, while three-point correlations can give rise to L patterns with various orientations. Our synthetic dataset comprises 2000 training images, 1000 validation images, and 2000 testing images. All reported results are based on the testing set.

Table 1: Accuracy and parameters of CNN and HoCNN models on Texture classification

| Model | Accuracy (%) | # Params |
|---|---|---|
| CNN (baseline) | 59.14 | 492 |
| HoCNN (up to 2$^{nd}$ order) | 82.42 | 293 |
| HoCNN (up to 3$^{rd}$ order) | 89.02 | 488 |
| HoCNN (up to 4$^{th}$ order) | 92.32 | 1259 |

We framed the problem as an image classification task with 10 different classes corresponding to various N-point correlations (N ranging from 1 to 4). To establish a baseline, we first tested a CNN model consisting of 3 blocks (2 convolutional - 1$^{st}$ conv layer with 10, 2x2 kernels and 2$^{nd}$ conv layer with 2, 8x2 kernels; both followed by batch normalization, ReLu nonlinearity and max pooling and 1 fully connected layer). Our results indicate (see **Table** 1 and **Figure** 2) that this basic architecture fails to correctly discriminate among the different 2-point and 3-point correlations. We then repeated the classification task using three different HoCNNs with expansions up to the 2$^{nd}$, the 3$^{rd}$, and the 4$^{th}$ order kernels (2 kernels of size 2x2) and an additional convolutional (conv layer with 2 kernels of size 8x2 + batch norm. + ReLU + pooling) and fully connected layer replicating the structure of the baseline CNN. These networks produced increasingly good performance (**Table** 1 and **Figure** 2), highlighting the need for our proposed higher-order convolution approach. For completeness, we report the number of parameters of the different architectures in **Table** 1.

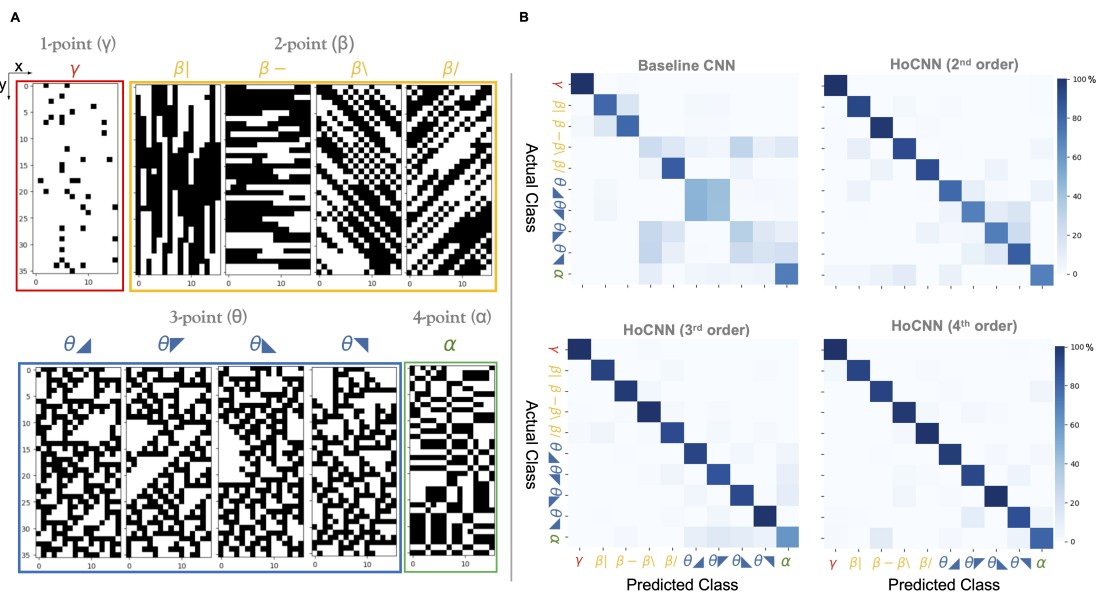

Figure 2: **Multipoint correlations and glider classification** (A) Textures generated with N-point gliders (N ranging from 1 to 4), totaling 10 classes when taking into account parity constraints. (B) Confusion matrices for different models, from top left: baseline CNN; top right: Higher-order CNN (HoCNN) with kernels expanded up to the 2$^{nd}$ order; bottom-left: HoCNN with kernels expanded up to the 3$^{rd}$ order; bottom-right: HoCNN with kernels expanded up to the 4$^{th}$ order. Taken together the four confusion matrices show that higher-orders progressively allow our network to properly capture relevant features for image classification.

## 3.2 BENCHMARK DATASETS

### 3.2.1 MNIST, FASHIONMNIST, CIFAR-10, CIFAR-100

To further validate our approach and ensure its generalizability, we conducted extensive experiments on four well-established benchmark datasets: MNIST, FashionMNIST, CIFAR-10, and CIFAR-100.

We focused on these relatively small datasets to allow for comprehensive testing and analysis. To ensure the robustness of our results and account for the inherent variability in neural network training, we conducted multiple independent runs for each experiment. Specifically, we initialized and trained each model with 50 different random seeds. This approach provides a more reliable estimate of model performance by mitigating the effects of random initializations.

We compared our HoCNN with $3^{rd}$ order kernels (3×3) against a standard CNN baseline with equivalent kernel size. Notably, extending to $4^{th}$ order kernels showed no significant improvement in test accuracy. Complete architectural and training details are provided in **Appendix** A.3, with parameter comparisons in **Table** 2. Test accuracy results, averaged over 50 realizations (**Figure** 3 and **Table** 2), demonstrate consistent performance advantages for HoCNN across all datasets, with particularly notable improvements on the more complex CIFAR-10 and CIFAR-100 tasks. These results suggest fundamentally different representational capabilities in HoCNN (further explored in **Section** 4). Additional experiments with a deeper and more complex CNN architecture, detailed in **Appendix** A.1, showed inferior performance compared to our simpler HoCNN, highlighting the efficiency of higher-order feature extraction.

**Training Details.** For all, experiments previously introduced experiments, we used AdamW optimizer (Loshchilov, 2017) with learning rate 0.001, weight decay 5e-4, batch size 64, and cross-entropy loss. Input images were normalized using z-score standardization, without any additional data augmentation.

### 3.2.2 IMAGENETTE

To investigate the scalability and real-world applicability of our approach, we conducted experiments with deeper architectures and a more challenging dataset. Specifically, we implemented Higher-order Convolution in a ResNet-18 (He et al., 2016) architecture (HoResNet-18) and evaluated it on Imagenette (Howard, 2020), a subsampled dataset from ImageNet containing 10 classes. The HoResNet-18 achieved better test accuracy compared to the standard ResNet-18 (see **Table** 2), demonstrating a performance improvement of over 1 percentage point on this challenging dataset.

The HoResNet-18 follows a similar structure to the standard ResNet-18, with a key modification in the first stage, where higher-order residual blocks are used. The remaining stages utilize standard residual blocks. This hybrid approach allows for the incorporation of higher-order convolutions while maintaining the overall structure of the ResNet architecture. The parameter count for both models is provided in **Table** 2, while architectural details are presented in the **Appendix** A.3.1. Additionally, our HoResNet-18 architecture achieves superior performance compared to a ResNet-18 adapted VOneNet (Dapello et al., 2020), a biologically-inspired model, while utilizing fewer parameters, as detailed in **Appendix** A.4.

These results provide strong evidence for the efficacy of our higher-order convolution approach, demonstrating successful scaling to deeper architectures and more complex datasets. By consistently outperforming the baseline across diverse benchmarks and multiple random initializations, our model exhibits robust improvements in image classification performance beyond simple tasks.

**Training Details.** For Imagenette specifically, images were normalized using mean = [0.5, 0.5, 0.5] and standard deviation = [0.5, 0.5, 0.5], in order to align with VOneNet (Dapello et al., 2020) preprocessing. All the other training details remain consistent with **Subsection** 3.2.1.

## 4 IMAGE STATISTICS SENSITIVITY AND NEURAL REPRESENTATIONS

To understand how our Higher-order Convolutional layer (HoConv) processes information differently from standard Convolutional (Conv) layers, we conducted two complementary analyses: an investigation of sensitivity to image statistics through perturbations (akin to adversarial attacks) and a study of representational geometry.

To directly quantify HoCNN's sensitivity to higher-order image statistics, we conducted a systematic perturbation analysis on the CIFAR-10 test dataset using both architectures (50 pretrained models each), while keeping the network trained on the unperturbed images. We generated synthetic textures with varying orders of statistical correlations (1-, 2-, 3-, and 4-point) and interpolated them with test images at different intensity (I) levels ($\text{img}_{perturbed} = (1\text{-I}) \times \text{img}_{CIFAR-10} + \text{I} \times \text{img}_{perturbation}$ ,

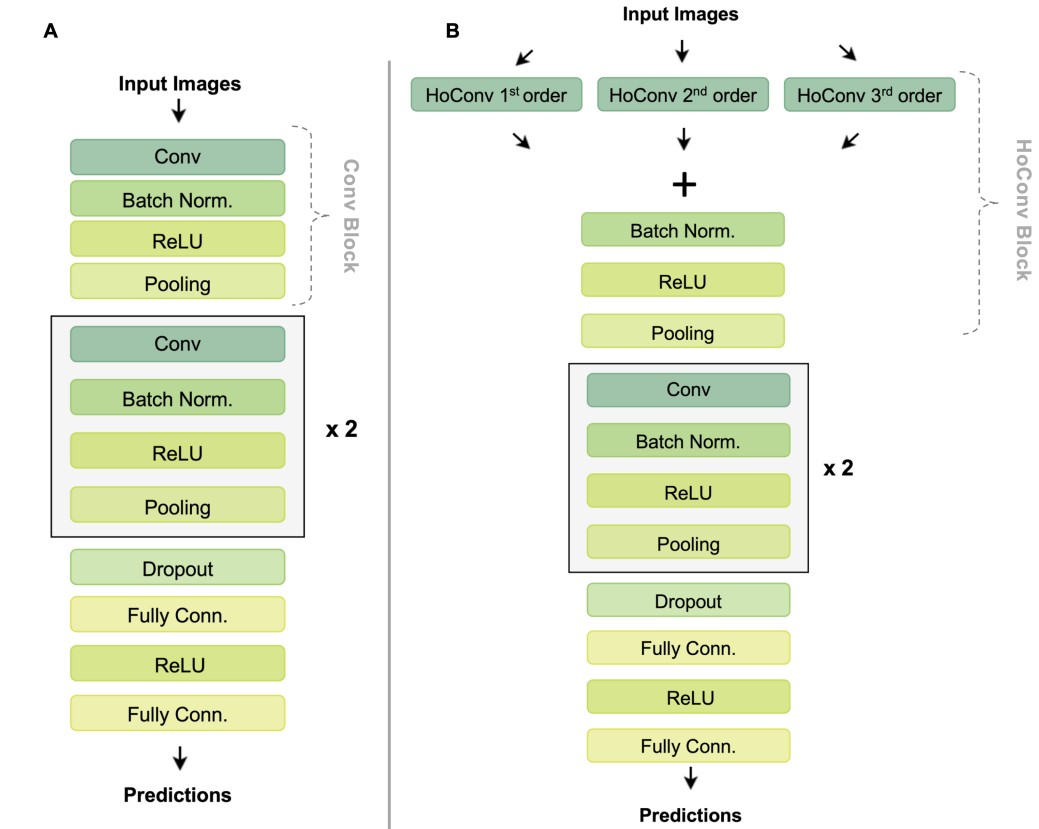

Figure 3: (A) Standard CNN architecture for Image Classification Benchmarks, the Conv Block represents the concatenation of Conv + Batch Norm. + ReLU + Pooling layers (B) Higher-Order CNN architecture: the first layer is split in three branches, representing the three orders of the Volterra expansion. After computing them separately, they're added together (mimicking once again the Volterra series expansion. Similarly, the HoConv Block represents the concatenation of HoConv + Batch Norm. + ReLU + Pooling layers. For both networks, the number of kernels depend on the classification task.

Table 2: Accuracy comparison (with $\pm\,\sigma$, standard deviation) of CNN and HoCNN models across image classification benchmarks, averaged over 50 seeds.

| Dataset | Accuracy (%) $\pm\,\sigma$ | | # Params | |
|---|---|---|---|---|
| | *CNN* | *HoCNN* | *CNN* | *HoCNN* |
| MNIST | $99.13 \pm 0.09$ | $99.30 \pm 0.07$ | 82,706 | 80,262 |
| FashionMNIST | $90.48 \pm 0.32$ | $90.95 \pm 0.35$ | 82,706 | 80,262 |
| CIFAR-10 | $69.76 \pm 0.64$ | $72.87 \pm 0.54$ | 82,706 | 80,262 |
| CIFAR-100 | $36.92 \pm 0.71$ | $40.42 \pm 0.84$ | 385,006 | 383,478 |
| Imagenette | 88.13 | 89.30 | 11,181,642 | 11,168,382 |

with $I \in [0.05, 0.20]$). While HoCNN achieves superior performance on CIFAR-10 test set images (see **Table** 2), it shows increased vulnerability to higher-order statistical perturbations. The performance degradation becomes progressively more pronounced as we move from lower to higher-order correlations, with the gap widening at higher perturbation intensities (see **Figure** 4 **A** and **B** for I = 0.12). For instance, with third-order perturbations at I = 0.12, HoCNN's accuracy drops by 78.0% compared to CNN's 73.7% (complete results in **Appendix** A.6).

This enhanced sensitivity to statistical perturbations suggests HoCNN's increased capacity to leverage higher-order patterns. To better understand how this statistical sensitivity translates into improved classification performance, we examined the model's internal representations using Representational Similarity Analysis (RSA) (Kriegeskorte et al., 2008) on CIFAR-10, analyzing activations from 100 test images (10 per class) averaged across 50 model realizations. The Conv and HoConv blocks exhibit distinct representational geometries (**Figure** 4A & B), with the latter showing more pronounced class-specific patterns. Analysis of pairwise distance distributions (**Figure** 4D) reveals that HoConv representations are significantly more dispersed in the high-dimensional space, indicating its ability to capture more diverse and discriminative features. Detailed analysis of individual order components and layer-wise correlations is presented in **Appendix** A.7.

Together, these analyses demonstrate that HoCNN's superior classification performance stems from its ability to capture and leverage higher-order statistical features in natural images, resulting in richer and more discriminative internal representations.

## 5 DISCUSSION

Our study introduces a novel approach to image classification by extending convolutional neural networks with higher-order learnable convolutions, inspired by complex nonlinear biological visual processing. Our findings indicate that expansion beyond the $4^{th}$ order is unnecessary, a conclusion supported by both theoretical insights and empirical evidence. Koenderink & van Doorn (2018) analysis of natural image statistics reveals that the quadric, cubic and quartic power dominates approximately 63%, 35%, and 2% of the pixels, respectively. Our experimental results align with this distribution, showing no significant performance gains beyond the $3^{rd}$ order in benchmark tasks and the $4^{th}$ in structured visual textures. Through systematic perturbation analysis, we validate this alignment by isolating the contributions of specific image statistics to model performance. While adversarial perturbations are typically employed to assess and improve network robustness (Szegedy, 2013; Goodfellow et al., 2014) or analyze network behavior (Carter et al., 2019; Santurkar et al., 2019), we leverage them to probe our model's sensitivity to distinct orders of image statistics, demonstrating how different orders of convolution process specific aspects of the input.

Our approach to structured visual textures relates our findings to current mechanistic models in computational neuroscience, particularly those focusing on the Drosophila visual system, which is known to be sensitive and selective to different 3-point correlations (Fitzgerald & Clark, 2015), (Clark et al., 2014). This connection strengthens the relevance of our model and highlights its potential to provide insight into visual processing in both biological and artificial systems.

The consistent outperformance of our Higher-order CNN (HoCNN) across various datasets demonstrates the effectiveness of incorporating these biologically inspired computations. Representational Similarity Analysis (RSA) reveals distinct geometries, providing evidence that our model processes visual information differently from standard CNNs, suggesting different strategies for encoding and processing visual information.

Future research directions include exploring deeper architectures with multiple higher-order convolutional layers, which presents challenges in balancing increasing nonlinearity across layers. To a certain extent, the scaling technique we employed (see **Section** 2) revealed better results compared to the case when it was not used. While computational complexity represents a current limitation, we envision future improvements through low-rank tensor decomposition methods. These techniques, which have proven successful in transformer architectures (Hu et al., 2021), could maintain the benefits of higher-order interactions while significantly reducing computational overhead.

In conclusion, our higher-order CNN represents a promising direction for more effective and biologically inspired computer vision models. By exploiting visual patterns and relationships more efficiently, even in shallow architectures, our approach opens new possibilities for advancing the field. Future research should focus on understanding the interplay between different orders of nonlinearity, especially for spatiotemporal inputs (i.e. videos), on exploring more complex architectures, and investigating potential applications beyond image classification, such as object detection or semantic segmentation.

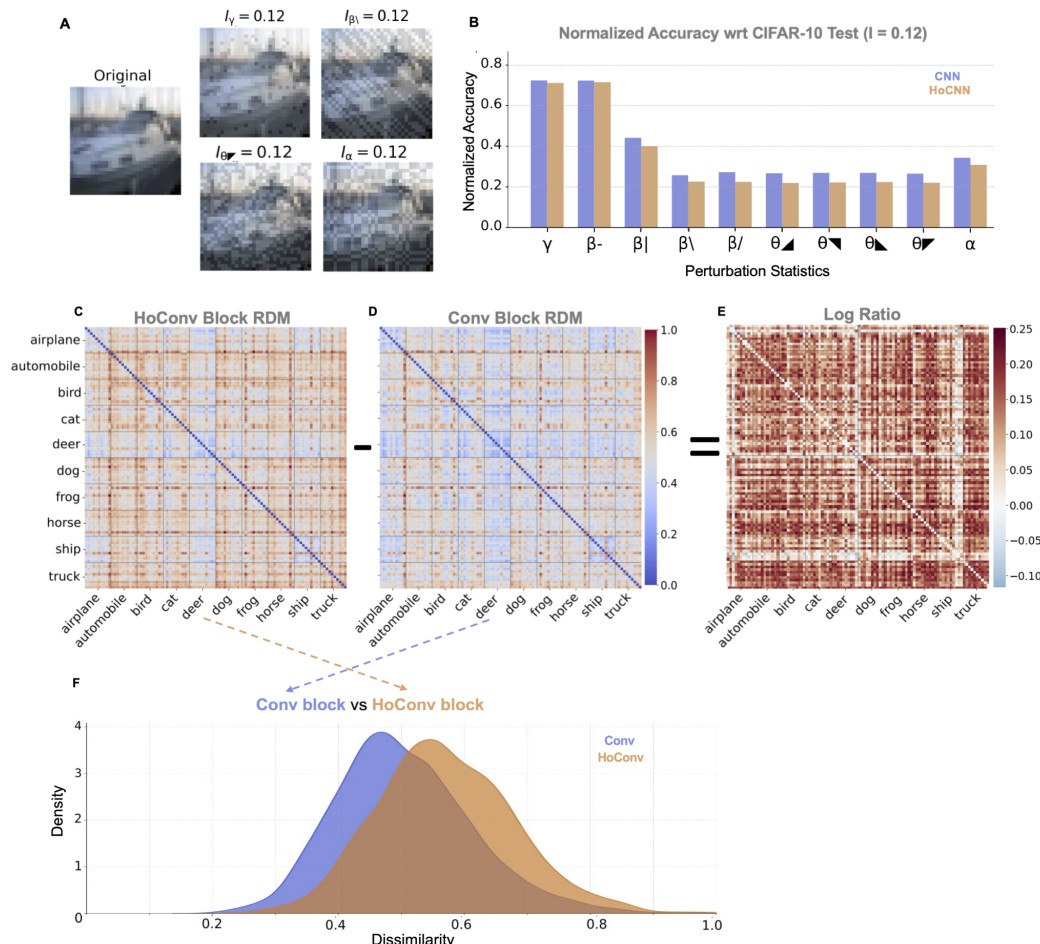

Figure 4: **Perturbation Analysis and Neural Representations**. (A) An exemplar test image (CIFAR-10) and perturbed examples with different 1, 2, 3, 4-point correlations statistics and common intensity (I = 0.12). (B) Normalized (wrt unperturbed CIFAR-10) accuracy for Convolutional (Blue) vs Higher-order (Orange) networks: performances of Higher-order network are systematically worse for more structured perturbations at fixed intensity (I = 0.12). (C) Representational Dissimilarity Matrix (RDM) for baseline Convolutional (Conv) Block (Conv layer, Batch Norm, ReLU, Pooling). (D) RDM for the Higher-order Convolutional (HoConv) block. (E) Log Ratio between the two RDMs, capturing different representational geometries between the two blocks. (F) An alternative view of the HoConv block effectiveness versus the Conv block, comparing the pairwise distance distribution across entries of each RDM. The orange distribution, representing the HoConv block, is significantly shifted to the right, supporting the enhanced image classification performances observed in **Figure** 5 and **Table** 2.

ACKNOWLEDGMENTS

In compliance with the double-blind review process, author information and specific acknowledgments have been omitted from this submission. A comprehensive acknowledgment section, as well as the code implementation, will be included upon acceptance of the paper.

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

# A  APPENDIX

## A.1  COMPUTATIONAL COSTS VERSUS TRADITIONAL METHODS

We analyze the computational requirements of HoCNN versus traditional CNNs. Our HoCNN architecture for MNIST, Fashion MNIST and CIFAR-10 consists of:

- First layer: $3 \times 3$ Volterra kernels (8 channels) with first, second, and third-order interactions
- Two standard convolutional blocks (16 and 32 channels)
- Two fully connected layers with dropout ($p = 0.25$)

Total parameters: 80,262

In order to provide a comparison with a more complex CNN model, we additionally tested a deeper version of the Baseline CNN model. In terms of architecture the Deep CNN network consist:

- Eight convolutional blocks ($3 \times 3$ kernels) with channel progression: $8 \rightarrow 8 \rightarrow 16 \rightarrow 16 \rightarrow 32 \rightarrow 32 \rightarrow 64 \rightarrow 64$
- Two fully connected layers with dropout ($p = 0.25$)

Total parameters: 108,130 (25% more than HoCNN)

Despite fewer parameters, HoCNN achieves superior accuracy (avg 72.35% vs 71.20%).

**Computational complexity scaling:**

- Standard convolution: $\mathcal{O}(C_{out} \times C_{in} \times K^2 \times H \times W)$
- Second-order kernels: $\mathcal{O}(C_{out} \times C_{in} \times K^4 \times H \times W)$
- Third-order kernels: $\mathcal{O}(C_{out} \times C_{in} \times K^6 \times H \times W)$

Where: $C_{out}$ = output channels, $C_{in}$ = input channels, $K$ = kernel size, $H, W$ = feature map height and width.

## A.2 PLACEMENT OF HIGHER-ORDER LAYER & EFFECTS OF LARGER KERNELS IN CNN

We conducted extensive experiments to investigate both the optimal placement of higher-order operations within the network hierarchy and the effects of increased kernel sizes in standard CNNs. These experiments provide important insights into the architectural choices made in our main study.

### A.2.1 EFFECT OF HIGHER-ORDER LAYER PLACEMENT

We tested the introduction of the higher-order block at different depths in the network, specifically at the second and third convolutional layers. Our results reveal a consistent pattern across datasets: performance gradually degrades as the higher-order block is placed deeper in the architecture. For CIFAR-10, accuracy decreases from 71.17 ± 0.51% when the higher-order block is at the second convolutional layer to 70.29 ± 0.49% at the third layer. Similarly, for CIFAR-100, accuracy drops from 37.27 ± 0.80% to 36.09 ± 0.81%.

This performance degradation can be attributed to earlier layers potentially capturing and overfitting to spurious correlations. While standard convolutions theoretically preserve higher-order correlations, the cascade of nonlinearities and pooling operations can introduce and amplify spurious correlations, potentially compromising the network's ability to learn meaningful representations at deeper layers.

### A.2.2 IMPACT OF KERNEL SIZE

We also investigated whether simply increasing kernel sizes in standard CNNs could achieve benefits similar to our higher-order operations. Despite larger kernels theoretically being capable of capturing higher-order correlations (as demonstrated by improved performance on our synthetic texture dataset), experiments with 5×5 and 7×7 kernels showed decreased test accuracy compared to our baseline CNN. For CIFAR-10, accuracy decreased to 68.72 ± 0.63% with 5×5 kernels and 67.51 ± 0.61% with 7×7 kernels. The pattern was similar for CIFAR-100, where accuracy dropped to 36.04 ± 0.76% with 5×5 kernels and 34.01 ± 0.72% with 7×7 kernels.

Our choice of 2×2 kernels for the main experiments was motivated by the generative process of our synthetic textures, which are based on gliders up to 4 pixels arranged in 2×2 patches. The results suggest that while larger kernels might have the capacity to capture higher-order interactions, the linear nature of standard convolutions makes it challenging to effectively learn these patterns during optimization.

### A.2.3 COMPUTATIONAL CONSIDERATIONS

The placement of higher-order operations in early layers offers computational advantages. In typical convolutional architectures, the number of channels increases with network depth. Introducing higher-order kernels in later layers, where channel count is larger, would substantially increase both parameter count and computational complexity. Our architectural choice of early placement thus provides an effective balance between computational efficiency and performance.

These findings support our design decision to introduce higher-order operations early in the network, where they can directly process raw image statistics before potential distortion by successive processing layers. The results demonstrate that explicitly modeling higher-order interactions through dedicated operations provides a more effective approach for capturing and leveraging complex statistical features compared to simply increasing kernel sizes in standard CNNs.

## A.3 ARCHITECTURES AND TRAINING PROCEDURE: MNIST, FASHIONMNIST, CIFAR-10, CIFAR-100 & IMAGENETTE

For image classification benchmarks, we employed a baseline CNN and a HoCNN architecture, with varying complexity depending on the dataset. For MNIST, FashionMNIST, and CIFAR-10, we used a simple architecture consisting of three convolutional blocks (Convolutional Layer, Batch Normalization, ReLU nonlinearity, and Max pooling) followed by two fully connected layers interleaved with dropout and ReLU nonlinearity (see **Figure** 3). For these datasets, we used 9, 18, and 36 kernels respectively for the convolutional layers of the baseline CNN, while the HoCNN used 8, 16, and 32 kernels. This configuration ensured that the baseline CNN had a comparable, slightly higher

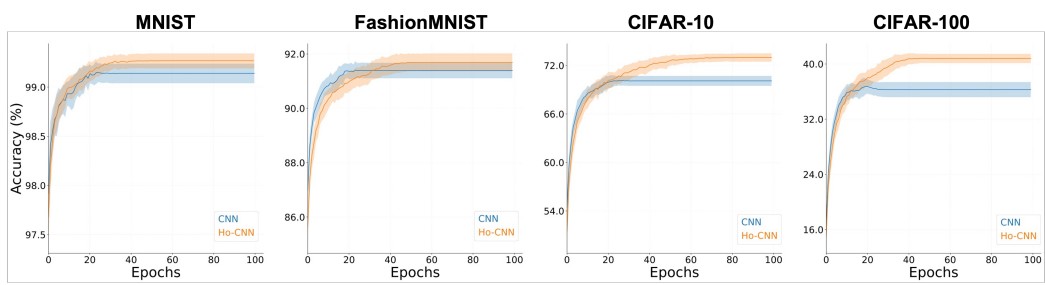

Figure 5: **Validation accuracy curves for CNN and HoCNN on image classification benchmarks**. Learning curves comparing the performance of the baseline CNN and the proposed HoCNN on MNIST, FashionMNIST, CIFAR-10, and CIFAR-100 datasets. The HoCNN consistently outperforms the CNN across all benchmarks, demonstrating the effectiveness of incorporating higher-order interactions in the convolutional layers.

number of parameters (82,706 vs. 80,262). For CIFAR-100, we increased the kernel counts to 34, 68, and 136 for the baseline CNN, and 32, 64, and 128 for the HoCNN, resulting in 385,006 and 383,478 parameters respectively.

The validation accuracy curves for the CNN and HoCNN on these benchmarks are provided in the **Figure** 5, further illustrating the consistent performance advantage of the HoCNN across datasets.

Our training strategy utilized the AdamW optimizer with weight decay and learning rate scheduling (ReduceLROnPlateau). We employed Early Stopping with patience for model selection. For MNIST, FashionMNIST, CIFAR-10, and CIFAR-100, we trained 50 model realizations with different random seeds to ensure reliable performance assessment. For Imagenette, we conducted a single comprehensive evaluation run.

### A.3.1 ARCHITECTURES AND PARAMETER COUNTS: IMAGENETTE

Standard ResNet-18 architecture, totaling 11,181,642 parameters:

Initial Block:

- Convolutional layer ($7 \times 7$ kernels, 64 channels)
- Batch normalization
- ReLU activation
- Max pooling ($3 \times 3$)

Four main stages, each containing 2 residual blocks:

- Stage 1: 64 channels (2 blocks)
- Stage 2: 128 channels (2 blocks)
- Stage 3: 256 channels (2 blocks)
- Stage 4: 512 channels (2 blocks)

Final layers:

- Global average pooling
- Fully connected layer to output classes

Higher-Order ResNet-18 (HoResNet-18), totaling 11,168,382 parameters:

Initial Block (similar to ResNet-18):

- Convolutional layer ($7 \times 7$ kernels, 30 channels)
- Batch normalization

- ReLU activation
- Max pooling ($3 \times 3$)

Four main stages, with a hybrid approach:

- Stage 1: 30 channels with higher-order residual blocks (2 blocks)
- Stage 2: 128 channels with standard residual blocks (2 blocks)
- Stage 3: 256 channels with standard residual blocks (2 blocks)
- Stage 4: 512 channels with standard residual blocks (2 blocks)

Final layers (similar to ResNet-18):

- Global average pooling
- Fully connected layer to output classes

### A.3.2 TRAINING SETUP: IMAGENETTE

Data Preprocessing:

- Training images: Random resized crop to $224 \times 224$, random horizontal flip, normalization
- Test images: Resize to $256 \times 256$, center crop to $224 \times 224$, normalization
- Both models use normalization with mean and std of [0.5, 0.5, 0.5] which results in rescaled pixels with mean close to zero ([-0.07, -0.09, -0.15]) and std close to 0.5 ([0.55, 0.54, 0.58])

Training Configuration:

- Batch size: 64
- Loss function: Cross-entropy
- Optimizer: AdamW with learning rate 0.001 and weight decay 5e-4
- Learning rate scheduling: ReduceLROnPlateau (halves LR after 5 epochs without improvement)
- Early stopping: Implemented with 12 epochs patience

### A.4 COMPARISON WITH VONENET

We compared the performance of our HoResNet-18 architecture with a ResNet-18 adapted VOneNet (Dapello et al., 2020), a biologically-inspired model designed to better capture the properties of the visual cortex. Our HoResNet-18 model achieved superior accuracy on the Imagenette dataset (89.30%) compared to the VOneNet model (88.02%), while utilizing fewer parameters (11,168,382 vs. 14,445,322).

This improved performance aligns with the findings reported by the VOneNet authors, who observed lower ImageNet accuracy compared to standard ResNet architectures, despite achieving higher brain-scores (i.e., higher explained variance of neural activity recorded from specific brain areas in the visual cortex). This trade-off between ImageNet performance and brain-score can be verified on the Brain-Score framework website by sorting models based on their ImageNet top-1 accuracy.

Our results demonstrate that the HoResNet-18 architecture, equipped with higher-order convolutions, can outperform biologically-inspired models like VOneNet on standard image classification tasks while maintaining a more compact parameter footprint.

### A.5 TIED-WEIGHTS ISSUE

A depiction of how Higher-order terms go beyond simple non-pointwise nonlinearities is provided in **Figure** 6 while PCA analysis with different nonlinearities is provided in **Figure** 7, obtaining similar qualitative results as with ReLU.

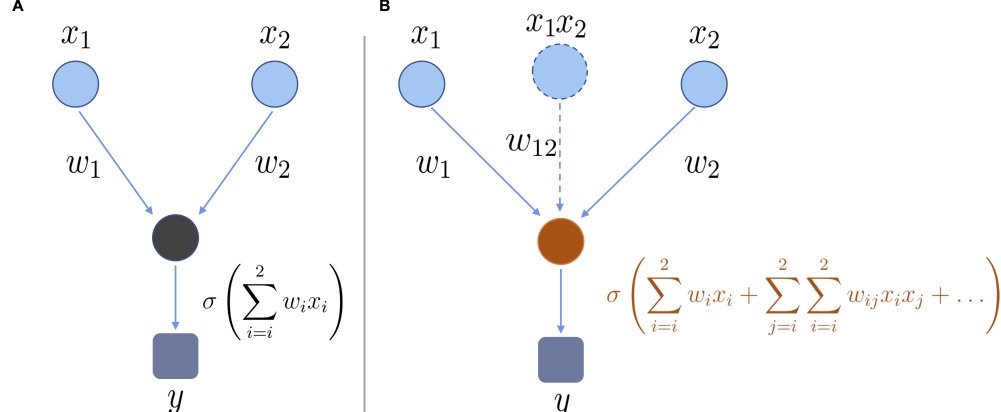

Figure 6: **Beyond pointwise nonlinearities** (A) Classical pointwise nonlinearity: the nonlinear function is applied after summing the input from the previous layer. A Taylor expansion of this non-linearity reveals the *tied-weight* problem as addressed in **Subsection** 1.2. (B) Non-pointwise nonlinearity includes the summation of quadratic or higher-order terms. By modulating the interactions between terms we can untie weights in the Taylor expansion: the optimization process is not saddled with discovering an efficient trade-off between good approximation and balance of different terms.

## A.6 SENSITIVITY TO IMAGE STATISTICS

More detailed results are presented in **Table** and (in \*\*bold\*\* largest decrease in performance across models).

Table 3: CNN Model accuracy (%) for each perturbation type and for different intensities I, in parentheses the decrease in accuracy with respect to the case without perturbation (CIFAR-10 avg test accuracy = 69.76%).

| Perturbation | I=0.05 | I=0.09 | I=0.12 | I=0.16 | I=0.20 |
|---|---|---|---|---|---|
| $\gamma$ | 67.0 (**-4.0%**) | 59.9 (-14.1%) | 50.5 (-27.6%) | 41.4 (-40.7%) | 34.1 (-51.2%) |
| $\beta$- | 66.5 (-4.7%) | 59.7 (-14.5%) | 50.4 (-27.7%) | 40.2 (-42.3%) | 31.2 (-55.2%) |
| $\beta$\| | 57.8 (-17.1%) | 41.5 (-40.5%) | 30.8 (-55.8%) | 24.6 (-64.7%) | 21.0 (-69.9%) |
| $\beta\backslash$ | 55.5 (**-20.5%**) | 31.2 (-55.2%) | 18.0 (-74.2%) | 13.2 (-81.0%) | 11.8 (-83.1%) |
| $\beta/$ | 56.6 (-18.9%) | 32.6 (-53.2%) | 19.0 (-72.8%) | 13.9 (-80.1%) | 12.2 (-82.5%) |
| $\theta_\llcorner$ | 54.2 (-22.2%) | 30.9 (-55.6%) | 18.6 (-73.3%) | 13.8 (-80.2%) | 11.9 (-82.9%) |
| $\theta_\ulcorner$ | 54.4 (-22.0%) | 31.2 (-55.3%) | 18.7 (-73.1%) | 13.9 (-80.1%) | 11.9 (-82.9%) |
| $\theta_\urcorner$ | 54.3 (-22.1%) | 31.0 (-55.6%) | 18.8 (-73.1%) | 13.9 (-80.0%) | 12.0 (-82.7%) |
| $\theta_\lrcorner$ | 54.2 (-22.3%) | 30.7 (-55.9%) | 18.5 (-73.5%) | 13.7 (-80.3%) | 11.9 (-83.0%) |
| $\alpha$ | 56.1 (-19.6%) | 36.0 (-48.4%) | 24.0 (-65.7%) | 18.3 (-73.7%) | 15.8 (-77.3%) |

## A.7 NEURAL REPRESENTATIONS: REPRESENTATIONAL SIMILARITY ANALYSIS

To gain deeper insight into how our Higher-order Convolutional layer (HoConv) processes information differently from a standard Convolutional (Conv) layer, we employed Representational Similarity Analysis (RSA) (Kriegeskorte et al., 2008) on the CIFAR-10 dataset. We analyzed activations from 100 test images (10 per class) averaged across 50 model realizations to mitigate the effects of random initialization (Ding et al., 2021). Our findings reveal distinct representational geometries between the two models. In **Figure** 8 we consider an alternative distance (hellinger) between the two RDMs, based on the consideration of the fact that values are between 0 and 1.

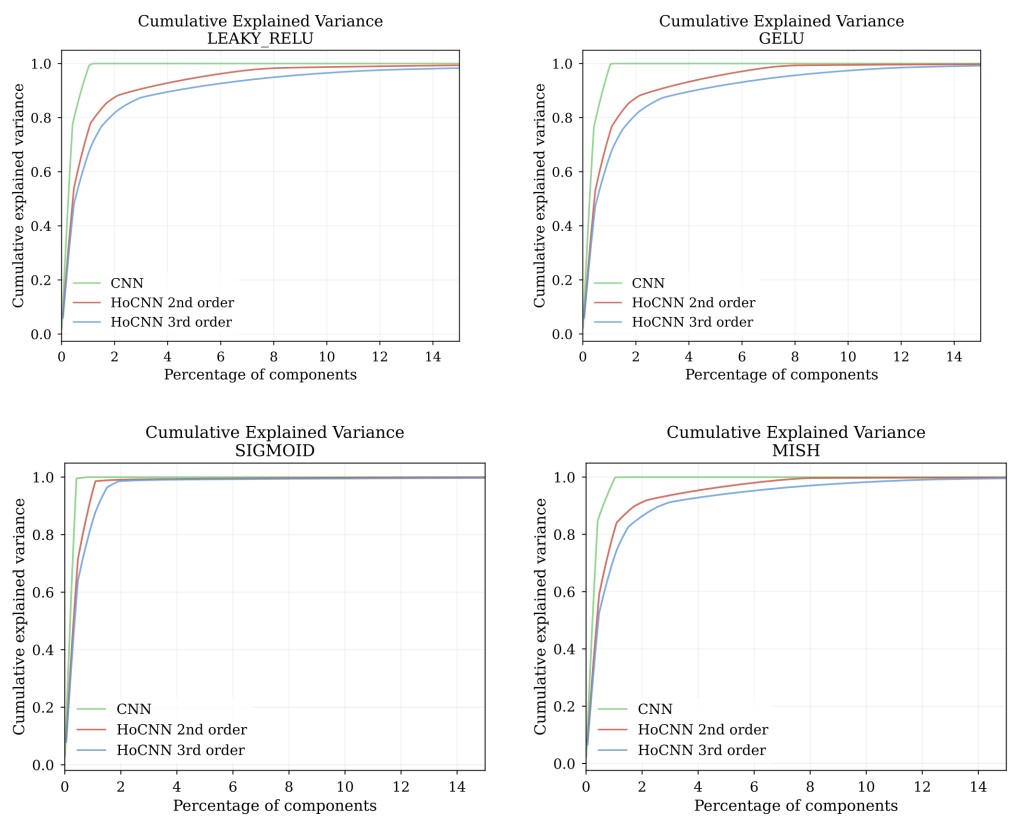

Figure 7: **PCA analysis with different nonlinearities: Leaky Relu, Gelu, Sigmoid, Mish**

Table 4: HO-CNN Model accuracy (%) for each perturbation type and for different intensities I, in parentheses the decrease in accuracy with respect to the case without perturbation (CIFAR-10 avg test accuracy = 72.87%).

| Perturbation | I=0.05 | I=0.09 | I=0.12 | I=0.16 | I=0.20 |
|---|---|---|---|---|---|
| $\gamma$ | 70.2 (-3.6%) | 62.5 (**-14.2%**) | 51.9 (**-28.8%**) | 41.2 (**-43.4%**) | 32.8 (**-55.0%**) |
| $\beta$- | 69.3 (**-4.9%**) | 61.7 (**-15.3%**) | 52.1 (**-28.5%**) | 41.3 (**-43.3%**) | 31.7 (**-56.5%**) |
| $\beta$\| | 58.5 (**-19.8%**) | 40.3 (**-44.8%**) | 29.1 (**-60.0%**) | 23.2 (**-68.2%**) | 20.2 (**-72.3%**) |
| $\beta\backslash$ | 58.5 (-19.7%) | 30.6 (**-58.0%**) | 16.5 (**-77.4%**) | 12.6 (**-82.7%**) | 11.7 (**-84.0%**) |
| $\beta/$ | 58.8 (**-19.3%**) | 30.5 (**-58.2%**) | 16.4 (**-77.5%**) | 12.5 (**-82.8%**) | 11.7 (**-83.9%**) |
| $\theta_{\llcorner}$ | 56.3 (**-22.7%**) | 29.1 (**-60.1%**) | 16.0 (**-78.0%**) | 12.1 (**-83.4%**) | 10.9 (**-85.0%**) |
| $\theta_{\ulcorner}$ | 56.3 (**-22.8%**) | 29.1 (**-60.0%**) | 16.2 (**-77.8%**) | 12.2 (**-83.2%**) | 11.0 (**-84.9%**) |
| $\theta_{\urcorner}$ | 56.4 (**-22.6%**) | 29.4 (**-59.6%**) | 16.4 (**-77.5%**) | 12.3 (**-83.1%**) | 11.0 (**-84.9%**) |
| $\theta_{\lrcorner}$ | 56.3 (**-22.8%**) | 29.1 (**-60.1%**) | 16.1 (**-77.9%**) | 12.2 (**-83.3%**) | 11.0 (**-85.0%**) |
| $\alpha$ | 58.4 (**-19.8%**) | 35.8 (**-50.9%**) | 22.5 (**-69.2%**) | 17.0 (**-76.6%**) | 14.9 (**-79.6%**) |

### A.7.1 ORDER-WISE REPRESENTATIONAL DIFFERENCES

The 2nd and 3rd order components of the HoConv layer can be analyzed independently and extract different representations, with progressively smaller mean dissimilarity compared to the baseline Conv layer (**Figure** 11). These order-wise differences contribute to the overall representational differences between CNN and HoCNN.

Our analysis of neural representations revealed distinct patterns across different order components (see **Figure** 9 and **Figure** 11). The standard convolutional layer exhibited a unimodal distribution of pairwise distances, indicating a consistent representation of stimuli. In contrast, the 2nd order

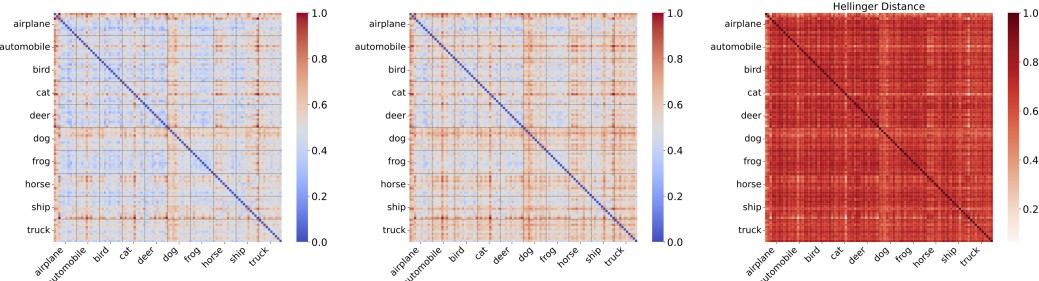

Figure 8: (Left) Representational Dissimilarity Matrix (RDM) for baseline Convolutional (Conv) Block (Conv layer, Batch Norm, ReLU, Pooling). (Middle) RDM for the Higher-order Convolutional (HoConv) block. (Right) Hellinger Distance between the two RDMs, capturing different representational geometries between the two blocks

component of the HoCNN showed a bimodal distribution, suggesting two distinct scales of representation. This bimodality could indicate the model's ability to capture both fine-grained similarities and broader categorical differences between stimuli.

The $3^{rd}$ order component demonstrated a wider spread of distances, with a peak at lower dissimilarity values. This pattern suggests that the $3^{rd}$ order component might be capturing more subtle relationships or features among the stimuli, complementing the representations of the lower-order components.

These findings point to a more nuanced and hierarchical representation in HoCNNs compared to standard CNNs. The multi-order structure appears to enable the model to simultaneously process information at various scales and levels of abstraction. This could potentially explain the enhanced performance of HoCNNs on certain tasks, as they may be better equipped to capture complex, multi-scale patterns in the input data.

However, further investigation is needed to fully understand the underlying mechanisms and implications of these representational differences. Future work could explore how these distinct representations contribute to specific aspects of model performance, and whether they align with known principles of biological visual processing.

### A.7.2 LAYER-WISE REPRESENTATIONAL DIFFERENCES

The representational difference between CNN and HoCNN becomes more explicit when all orders are summed together and fed to the batch normalization, ReLU, and max pooling layers (see **Figure** 3), giving rise to the output of our HoConv block. We quantify these differences using Representational Dissimilarity Matrices (RDMs) on a subset of the testing set (**Figure** 4). We compare blocks instead of single layers because the standard convolution is linear, and equipping it with a nonlinearity layer makes the comparison fairer. The Conv and HoConv RDMs show clear structural differences (**Figure** 4A  B), with the HoConv exhibiting more pronounced class-specific patterns.

### A.7.3 CORRELATION BETWEEN RDMs ACROSS LAYERS

We further analyzed the average correlation between RDMs across layers of the CNN and HoCNN (**Figure** 10). The results confirm the divergence in representational structure, with correlations decreasing in higher-order layers. This suggests that the HoConv layer progressively captures more diverse and complex features compared to the standard Conv layer.

### A.7.4 RSA ON TEXTURE CLASSIFICATION

We extended the analysis presented in paragraph 4 of the main paper to the four models trained on synthetic texture data. This analysis aims to highlight the contributions of different kernel orders in texture classification. Following a similar methodology, we randomly selected 10 samples for each texture class from the test set. We then extracted activations at specific layers of the models

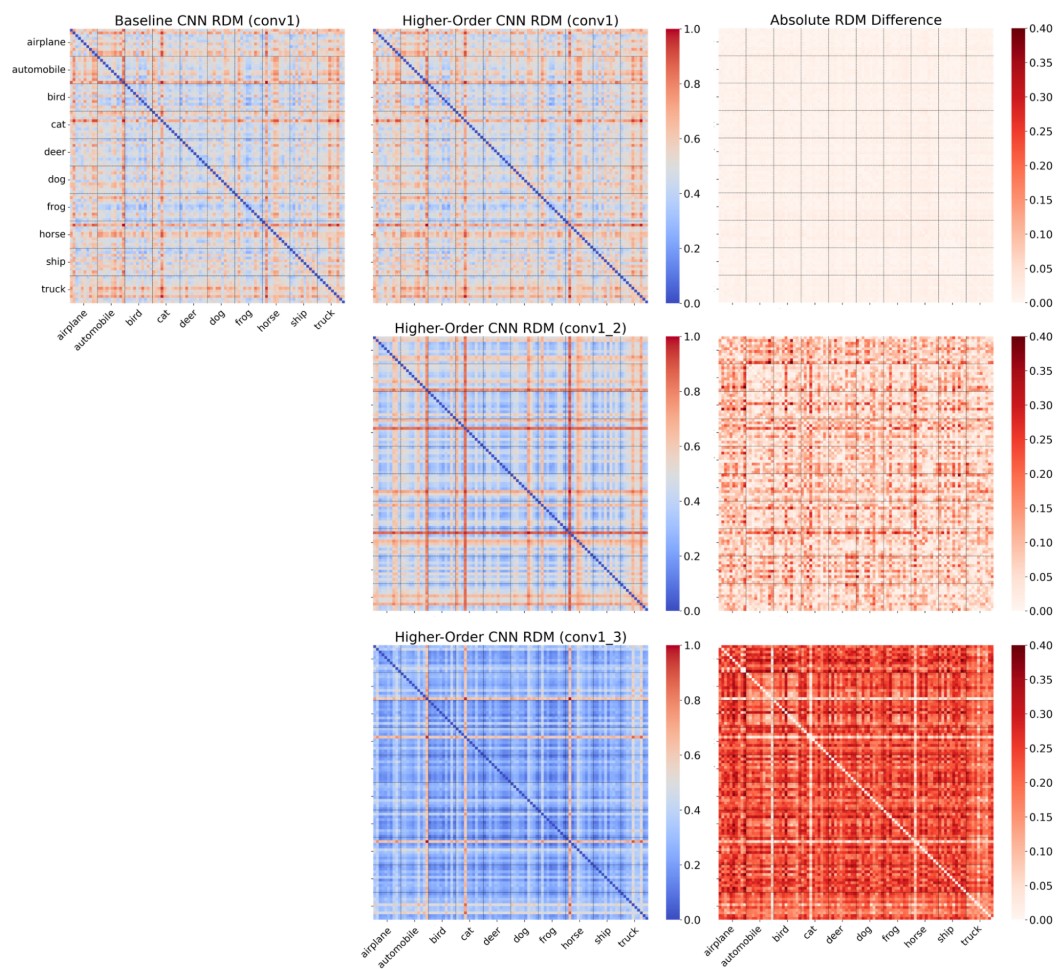

Figure 9: **Representational dissimilarity matrices across layers**. (First row) Representational Dissimilarity Matrices for (left) standard Conv layer (1$^{st}$ layer of the CNN), (middle) 1$^{st}$ order Ho-Conv layer, (right) absolute difference between the two RDMs. (Second row), the RDM for standard Conv is not repeated while (in the middle) we show the RDM for the activations extracted after the 2$^{nd}$ order HoConv layer; (right) the absolute difference between the standard Conv and the 2$^{nd}$ order HoConv RDMs, which presents different representational geometries.

and used these to construct representational dissimilarity matrices (RDMs), see **Figure** 12 , **Figure** 13and **Figure** 14.

These RDMs provide a visual representation of how the models distinguish between different texture classes at various processing stages. By comparing the RDMs across different layers and between the baseline CNN and HoCNN models, we can gain insights into how higher-order kernels influence the representation and discrimination of texture patterns.

The following figures present these RDMs, allowing for a comparative analysis of how information is processed and transformed through the network layers in both the standard CNN and the HoCNN architectures when dealing with synthetic texture data. This analysis complements our findings from natural image datasets and offers a more comprehensive view of the role of higher-order computations in visual processing tasks.

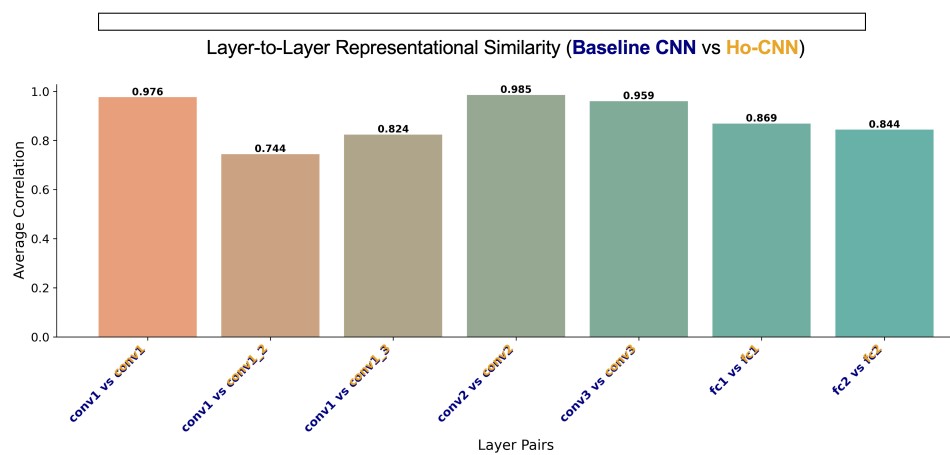

Figure 10: **Average Correlation between RDMs for matching layers (Baseline CNN vs HoCNN)**.

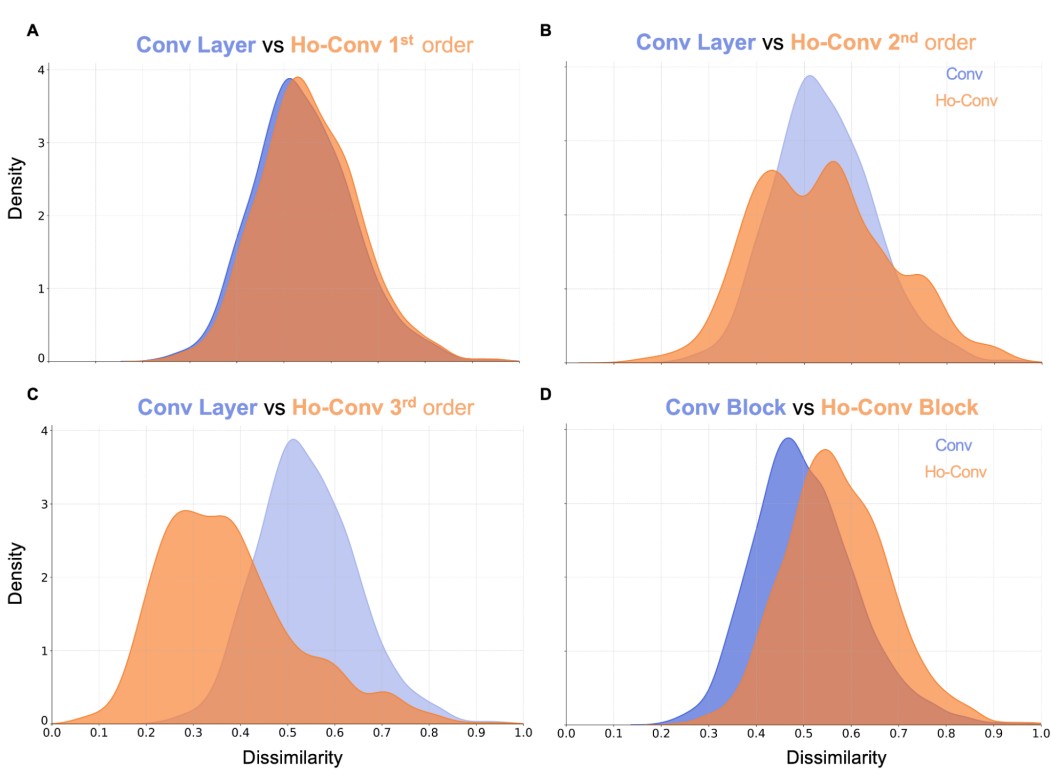

Figure 11: **Pairwise distance distribution of RDMs (CIFAR-10)**. An alternative way to represent RDM matrices is to plot the pairwise distance distribution of their entries. (A) We can see that at the 1st order the representations of the standard Conv Layer (here before the nonlinearity) and of the HoConv Layer are practically the same (1st order HoConv Layer is mathematically equivalent to a Conv Layer). (B) Representations for the 2nd order HoConv Layer are different from the standard convolutional layer: by themselves alone they don't bring better representations in terms of dissimilarity if compared with the standard Conv Layer ones. (C ) Similar considerations for 3rd order HoConv Layer representations. (D) When considering the output of the two blocks indeed (same as Fig 5. D), the situation is different: the distribution corresponding to the HoConv representations is considerably shifted, leading to "more" dissimilar representations on average.

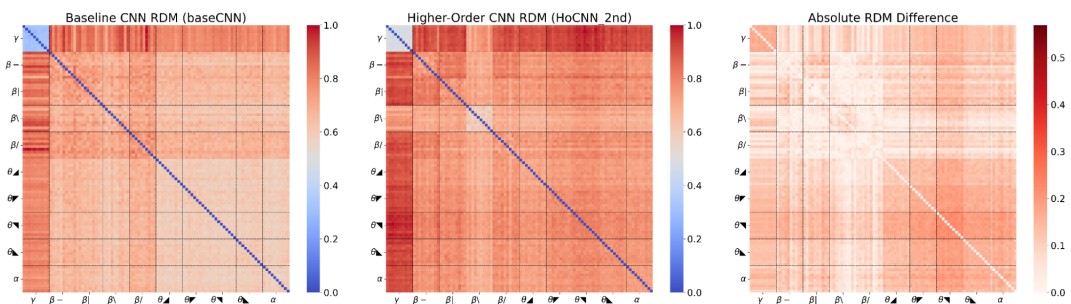

Figure 12: RDM comparison between baseline CNN and HoCNN 2nd order (summed 1st and 2nd order kernel), on the right absolute difference to highlight representational differences.

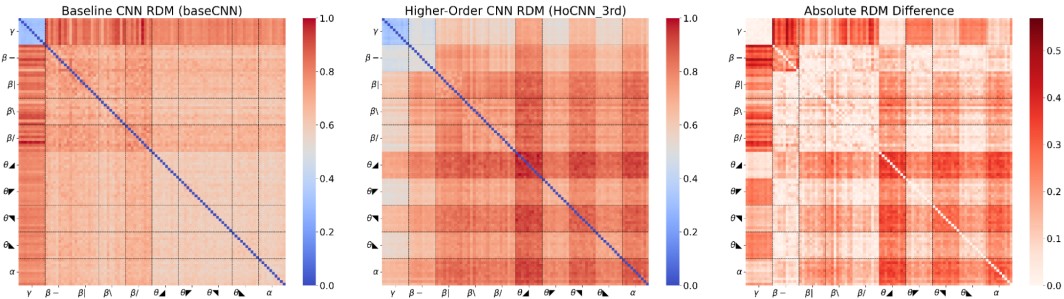

Figure 13: RDM comparison between baseline CNN and HoCNN 3rd order (summed 1st, 2nd and 3rd order kernel), on the right absolute difference to highlight representational differences.

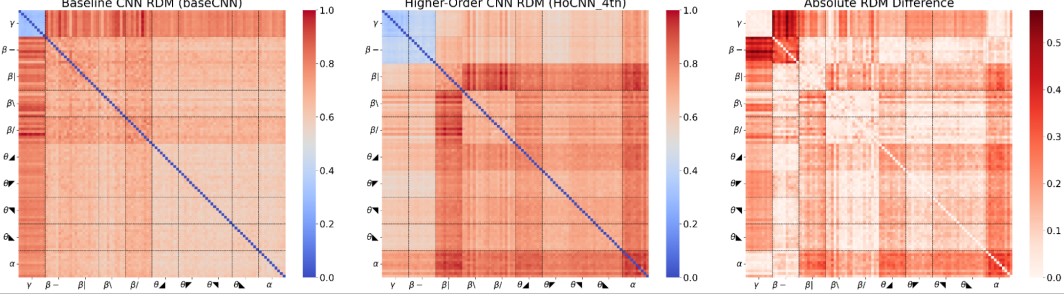

Figure 14: RDM comparison between baseline CNN and HoCNN 4th order (summed 1st, 2nd, 3rd and 4th order kernel), on the right absolute difference to highlight representational differences.

