# OpenReview forum: "Convolution goes higher-order: a biologically inspired mechanism empowers image classification."
_ICLR.cc/2025/Conference — Submitted to ICLR 2025_

### Official Review · Reviewer_ELAH · 2024-10-25

**Soundness:** 2
**Presentation:** 1
**Contribution:** 2
**Rating:** 6
**Confidence:** 3

**Summary:**

This paper introduces higher order convolutions in CNNs, citing the importance of higher-order statistics from neuroscience literature. The authors report favorable results in MNIST, F-MNIST, CIFAR-10 and CIFAR100.

**Strengths:**

The concept is rigorously introduced (I did not know what "Volterra-like" really meant but the equations were clear).

**Weaknesses:**

I apologize in advance for my short review, but the paper is actually too light on content for me to have too many opinions. The introductory text goes to page 3, there are a bunch of equations with summations taking up essentially a page, gigantic figures 1 and 2 that basically re-explains the equations, and just like that we are already at page 6. Then, in the experiments section, there are gigantic figures 4 and 5 describing architecture and validation accuracy taking up a whole page. This leaves only 3 pages of real results that I am reviewing. Nonetheless, here are the issues I have with this work:

- The models and datasets are too small to claim improvement over regular CNNs. Even the smallest CNNs widely accepted in literature are trained on ImageNet, and have parameter count in the millions. To be perfectly clear, I am open to small toy networks that illustrate mechanisms or specific interpretations, but that is not the case here. The models here are so small that residual connections (found in basically every "traditional CNN") are not even used, yet they claim to "outperform traditional CNN baselines" as quoted in the abstract. I hope the authors do not think I am gatekeeping computational resources but that their claims are overblown for the parameter count and dataset they are using, especially when there are no indications of this would work when scaled up.

- There is nothing in the paper about image preprocessing, image augmentations, loss functions, learning rates, dropout probabilities and everything else that should be properly controlled for (and more importantly, stated in the paper). As a reviewer this once again leaves me with doubts but nothing concrete to pinpoint.

**Questions:**

See weaknesses.

---

> ### Author Response · Authors · 2024-11-24
>
> Dear Reviewer ELAH,
>
> We appreciate your feedback regarding the paper's content density and space utilization. We revised our work to address these concerns. Firstly, we have re-organized the figures to optimize space: Figures 2 and 5 have been moved to the appendix.
>
> Secondly, we have enriched the content with:
>
> * Extended experimental results, including our new HoResNet-18 implementation
> * Added a more extensive description of training conditions
> * Complemented RDM with a direct quantification of HoCNN’s higher-order statistical sensitivity to structured perturbations (adversarial attack)
> * Quantitative perspective on tied-weight issue, to complement Section 2.1,  as noted by reviewer EyLg
>
> Here we address your concerns in a detailed way while we are updating the paper accordingly.
>
> ---
> **Scalability and Real-World Applicability**
>
> While we acknowledge the reviewer's valid point about the initial experimental scope, we have now conducted additional experiments with deeper architectures and more challenging datasets. Specifically, we implemented HoConv in a ResNet-18 architecture (HoResNet-18) and evaluated it on  [Imagenette](https://github.com/fastai/imagenette) (a subsampled dataset from ImageNet, containing 10 classes), achieving 89.30% test accuracy compared to 88.13% for standard ResNet-18. These results demonstrate that our approach successfully scales to deeper architectures and more complex datasets, addressing the concern about real-world applicability. This performance improvement of over 1 percentage point on a challenging dataset further validates the effectiveness of our approach beyond simple toy examples.
>
> We would also like to note that while ImageNet evaluation is indeed a gold standard, there are numerous recent works in major conferences, including ICLR (e.g., [Sanborn et al (2023)](https://arxiv.org/pdf/2209.03416), [Cohen & Welling (2016)](https://arxiv.org/pdf/1602.07576), [Benton et al (2020)](​​https://arxiv.org/pdf/2010.11882)), that validate novel architectural innovations on smaller-scale datasets (across domains, even the outstanding [“Attention is all you need”](https://proceedings.neurips.cc/paper_files/paper/2017/file/3f5ee243547dee91fbd053c1c4a845aa-Paper.pdf) Transformer paper doesn’t present scaling to large datasets). Nevertheless, we take the reviewer's point about scalability seriously and have now demonstrated our method's effectiveness on Imagenette with a full-scale ResNet architecture (11M+ parameters).
>
> For completeness, here we report architectural details and parameter count:
>
> 1. Standard ResNet-18 architecture totalling 11181642 parameters
>
> * Initial Block:
>     * A convolutional layer (7×7 kernels, 64 channels)
>     * Batch normalization
>     * ReLU activation
>     * Max pooling (3×3)
>
> * Four main stages, each containing 2 residual blocks:
>     * Stage 1: 64 channels (2 blocks)
>     * Stage 2: 128 channels (2 blocks)
>     * Stage 3: 256 channels (2 blocks)
>     * Stage 4: 512 channels (2 blocks)
>
> * Final layers:
>
>     * Global average pooling
>     * Fully connected layer to output classes
>
> 2. Higher-order Resnet-18 (Ho-Resnet-18), totalling 11168382 parameters. It follows a similar structure wrt Resnet-18 with a key modification in **bold**
>
> * Initial Block (similar to ResNet-18):
>     * Convolutional layer (7×7 kernels, **30 channels**)
>     * Batch normalization
>     * ReLU activation
>     * Max pooling (3×3)
>
> * Four main stages, but with a hybrid approach:
>     * **Stage 1: 30 channels with higher-order residual blocks (2 blocks)**
>     * Stage 2: 128 channels with standard residual blocks (2 blocks)
>     * Stage 3: 256 channels with standard residual blocks (2 blocks)
>     * Stage 4: 512 channels with standard residual blocks (2 blocks)
>
> * Final layers (similar to ResNet-18):
>     * Global average pooling
>     * Fully connected layer to output classes
>
> Additionally, the training setup for our experiments can be summarized as follows:
>
> * Data Preprocessing:
>     * *Training images*: Random resized crop to 224×224, random horizontal flip, normalization with mean and std of [0.5, 0.5, 0.5]
>     * *Test images*: Resize to 256×256, center crop to 224×224, normalization with mean and std of [0.5, 0.5, 0.5]
>
> * Training Configuration:
>     * Batch size: 64
>     * Loss function: Cross-entropy
>     * Optimizer: AdamW with learning rate 0.001 and weight decay 5e-4
>     * Learning rate scheduling: ReduceLROnPlateau (halves LR after 5 epochs without improvement)
>     * Early stopping: Implemented with 12 epochs patience
>
> We are now adding the analysis of this complex dataset in the revised paper.

---

> ### Author Response · Authors · 2024-11-24
>
> **Training Protocol and Reproducibility**
>
> We appreciate the reviewer's point about the need for detailed training specifications. In our initial experiments with MNIST, FashionMNIST, CIFAR10, and CIFAR100, we deliberately maintained minimal preprocessing (normalizing images to mean = [0.5] and std = [0.5]) and avoided augmentations to ensure a clear assessment of the models' capacity to capture higher-order statistics (see paragraph below for extended experiments on this, as also noted by Reviewer EyLg). This design choice was intentional, as some augmentations could potentially alter the statistical structure of the images and confound our analysis of HoCNN's sensitivity to higher-order statistics compared to baseline CNNs.
>
> For completeness, here are the specific training details used across our experiments:
>
> * Loss Function: Cross-entropy loss
> * Optimizer: AdamW with learning rate 0.001 and weight decay 1e-4
> * Dropout Probability 0.25.
> * Early Stopping: Implemented with 12 epochs patience
> * Learning Rate Scheduling: ReduceLROnPlateau with Factor: 0.5 and Patience: 5 epochs
>
> We acknowledge that these details should have been included in the main paper for reproducibility, and we will add them in the revised version. We now report extended experiments to address your point on the lightness of content.
>
> ---
> **Addressing HoCNN's Higher-Order Statistical Sensitivity**
>
> To strengthen our submission, as also noted by reviewer EyLg,  we demonstrate how HoCNN better captures higher-order statistics through a systematic perturbation analysis that directly quantifies its sensitivity to higher-order image statistics.
>
> Our methodology employs adversarial attacks (perturbations) on the CIFAR-10 test dataset (10,000 images), using the 50 pretrained models for both the baseline CNN and HoCNN. We generated synthetic textures with varying orders of statistical correlations (1-, 2-, 3-, and 4-point, similar to our synthetic dataset) and added them to the original test images at different intensity levels (ranging from 0.05 to 0.20).
>
> While HoCNN outperforms CNN on clean images (CIFAR test accuracy (%): CNN: 69.76 ± 0.64%; HoCNN: 72.87 ± 0.54%, see manuscript), its performance drops more severely when images are perturbed with higher-order statistical textures (see tables below, in **bold** largest decrease in performance - accuracy (%) - across models).
>
> * CNN Model accuracy (%) for each perturbation type and for different intensities I, in parentheses the decrease in accuracy with respect to the case without perturbation  (CIFAR-10 avg test accuracy = 69.76 %):
>
> | Perturbation | I=0.05 | I=0.09 | I=0.12 | I=0.16 | I=0.20 |
> |-------|---------|---------|---------|---------|---------|
> | γ    | 67.0 (**-4.0%**) | 59.9 (-14.1%) | 50.5 (-27.6%) | 41.4 (-40.7%) | 34.1 (-51.2%) |
> | β-   | 66.5 (-4.7%) | 59.7 (-14.5%) | 50.4 (-27.7%) | 40.2 (-42.3%) | 31.2 (-55.2%) |
> | β\|  | 57.8 (-17.1%) | 41.5 (-40.5%) | 30.8 (-55.8%) | 24.6 (-64.7%) | 21.0 (-69.9%) |
> | β\\  | 55.5 (**-20.5%**) | 31.2 (-55.2%) | 18.0 (-74.2%) | 13.2 (-81.0%) | 11.8 (-83.1%) |
> | β/   | 56.6 (-18.9%) | 32.6 (-53.2%) | 19.0 (-72.8%) | 13.9 (-80.1%) | 12.2 (-82.5%) |
> | θ◢   | 54.2 (-22.2%) | 30.9 (-55.6%) | 18.6 (-73.3%) | 13.8 (-80.2%) | 11.9 (-82.9%) |
> | θ◤   | 54.4 (-22.0%) | 31.2 (-55.3%) | 18.7 (-73.1%) | 13.9 (-80.1%) | 11.9 (-82.9%) |
> | θ◥   | 54.3 (-22.1%) | 31.0 (-55.6%) | 18.8 (-73.1%) | 13.9 (-80.0%) | 12.0 (-82.7%) |
> | θ◣   | 54.2 (-22.3%) | 30.7 (-55.9%) | 18.5 (-73.5%) | 13.7 (-80.3%) | 11.9 (-83.0%) |
> | α    | 56.1 (-19.6%) | 36.0 (-48.4%) | 24.0 (-65.7%) | 18.3 (-73.7%) | 15.8 (-77.3%) |
>
> * HO-CNN Model (CIFAR-10 avg test accuracy = 72.87 %) :
>
> | Perturbation | I=0.05 | I=0.09 | I=0.12 | I=0.16 | I=0.20 |
> |-------|---------|---------|---------|---------|---------|
> | γ    | 70.2 (-3.6%) | 62.5 (**-14.2%**) | 51.9 (**-28.8%**) | 41.2 (**-43.4%**) | 32.8 (**-55.0%**) |
> | β-   | 69.3 (**-4.9%**) | 61.7 (**-15.3%**) | 52.1 (**-28.5%**) | 41.3 (**-43.3%**) | 31.7 (**-56.5%**) |
> | β\|  | 58.5 (**-19.8%**) | 40.3 (**-44.8%**) | 29.1 (**-60.0%**) | 23.2 (**-68.2%**) | 20.2 (**-72.3%**) |
> | β\\  | 58.5 (-19.7%) | 30.6 (**-58.0%**) | 16.5 (**-77.4%**) | 12.6 (**-82.7%**) | 11.7 (**-84.0%**) |
> | β/   | 58.8 (**-19.3%**) | 30.5 (**-58.2%**) | 16.4 (**-77.5%**) | 12.5 (**-82.8%**) | 11.7 (**-83.9%**) |
> | θ◢   | 56.3 (**-22.7%**) | 29.1 (**-60.1%**) | 16.0 (**-78.0%**) | 12.1 (**-83.4%**) | 10.9 (**-85.0%**) |
> | θ◤   | 56.3 (**-22.8%**) | 29.1 (**-60.0%**) | 16.2 (**-77.8%**) | 12.2 (**-83.2%**) | 11.0 (**-84.9%**) |
> | θ◥   | 56.4 (**-22.6%**) | 29.4 (**-59.6%**) | 16.4 (**-77.5%**) | 12.3 (**-83.1%**) | 11.0 (**-84.9%**) |
> | θ◣   | 56.3 (**-22.8%**) | 29.1 (**-60.1%**) | 16.1 (**-77.9%**) | 12.2 (**-83.3%**) | 11.0 (**-85.0%**) |
> | α    | 58.4 (**-19.8%**) | 35.8 (**-50.9%**) | 22.5 (**-69.2%**) | 17.0 (**-76.6%**) | 14.9 (**-79.6%**) |

---

> > ### Author Response · Authors · 2024-11-24
> >
> > This greater vulnerability to statistical perturbations increases as we move from lower to higher-order correlations, with the performance gap becoming particularly pronounced at higher perturbation intensities. This stronger degradation in performance provides strong evidence that HoCNN's superior performance on the CIFAR-10 test accuracy stems from its enhanced ability to leverage higher-order statistical features. We are adding this analysis along with a new figure to the paper that thoroughly documents these findings.
> >
> > **A Quantitative Perspective on the Tied-Weight Issue**
> >
> > An additional result we are adding, based on Reviewer EyLg's comment, and that we feel substantially empower the content of our submission, is a quantitative analysis demonstrating how HoCNN addresses the "tied-weight issue". We are adding this analysis to the main paper.
> >
> > Our analysis methodology:
> >
> > 1. We generated a fixed input image containing binary textures exhibiting all possible 1-, 2-, 3-, and 4-point correlations for 2×2 patches, corresponding to fixing the x_i terms in Equation (4) of the paper.
> > 2. We initialized the same model architecture 10,000 times and focused on the  activation patterns after the convolutional block (Conv/HoConv + BatchNorm + ReLU + Pooling), effectively generating many different combinations of the w_i terms.
> > 3. We performed PCA on those activations and measured the number of components needed to explain 95% of the variance
> >
> > Our results, using ReLU nonlinearity as in our main architecture, show that, for 95% explained variance:
> >
> > * Baseline CNN requires 87 components (out of 9610, 0.9% of the total)
> > * HoCNN with 2nd order expansion requires 102 components (out of 1922, 5.3 % of the total)
> > * HoCNN with 3rd order expansion requires 159 components (out of 1922; 8.3 % of the total)
> >
> > These results quantitatively demonstrate that higher-order convolutions introduce more independent weights, leading to a richer representation space with less weight tying. The increasing number of components needed for variance explanation and the higher effective dimensionality directly show how our approach addresses the tied-weight issue.
> >
> > Additionally, we conducted the same analysis with different nonlinearities (including Leaky ReLU, GELU, Sigmoid, and the more modern Mish), which showed consistent results, further supporting our conclusions. These results on additional nonlinearities will be included in the appendix.
> >
> > To conclude, all the content presented in this response will be added to the revised version of our paper.
> >
> > Best regards,

---

> > > ### Comment · Reviewer_ELAH · 2024-11-25
> > > **Response**
> > >
> > > I thank the authors for the additional experiments. This would have been great to have in the initial submission, because as a reviewer I would ask about the following (for which you would have had ample time to address):
> > >
> > > - Normalization during preprocessing: Why is 0.5 mean/std used for normalization? The authors are comparing to ResNet-18, which uses the much more standardized mean 0 std 1, which works extremely well with the batch-normalization inherent in ResNet-18.
> > >
> > > - AdamW as optimizer: Once again this is an uncommon choice, as the standard approach is to use plain SGD with step LR schedules. AdamW overfits on the training data (as you will undeniably see that your models do very well on the training set compared to the test set, especially for ResNet-18) and does not fairly put ResNet-18 as a benchmark.
> > >
> > > - How does lack of augmentations mean that "ensure a clear assessment of the models' capacity to capture higher-order statistics"? An image is identifiable through augmentations because the key statistics underlying the identity of the image is preserved. Given an image of a tower, if you shear it by a few degrees, it is still recognizable as a tower because of traits such as a roof, windows, brick walls etc. If this messes up second order statistics that your model captures, then it is not capturing useful information.
> > >
> > > I am open to further replies from the authors.

---

> > > > ### Author Response · Authors · 2024-11-29
> > > >
> > > > Thank you for your thoughtful questions. We would like to address each point:
> > > >
> > > > 1. We are sorry, our explanations were not sufficiently clear. We agree that z-scoring the pixels is the standard procedure, and this is what we have done throughout the paper (synthetic dataset, MNIST, FashionMNIST, CIFAR-10, and CIFAR-100). For Imagenette instead we rescaled the pixels by mean = [0.5,0.5,0.5] and std = [0.5, 0.5, 0.5] which results in rescaled pixels with mean close to zero ([-0.07,  -0.09, -0.15]) and std close to 0.5 ([0.55,  0.54, 0.58]). We applied this pseudo z-scoring in order to align with the  VOneNet model ([Dapello et al. 2020](https://proceedings.neurips.cc/paper/2020/file/98b17f068d5d9b7668e19fb8ae470841-Paper.pdf)), an additional comparison, requested by Reviewer Rcmj. VOneNet uses this normalization scheme, and we maintained consistency across comparisons. While different normalization schemes exist (e.g., [PyTorch's ImageNet defaults](https://github.com/pytorch/examples/blob/main/imagenet/main.py) is: mean = [0.485, 0.456, 0.406], std = [0.229, 0.224, 0.225], as you suggested), our rationale was to apply the same preprocessing to all models trained from scratch, ensuring fair comparison.
> > > >
> > > >
> > > > 2. Concerning the optimizer choice, our monitoring shows that AdamW did not lead to significant overfitting in our experiments. At early stopping, both models showed comparable training accuracies (ResNet-18: 89.70%, HoResNet-18: 89.57%) relative to the test performance (ResNet-18: 88.13%; HoResNet-18: 89.30%). While SGD with step scheduling is indeed common, AdamW has shown good performance across various architectures in literature (see [this resource for models using AdamW]( https://paperswithcode.com/method/adamw), including image classification models),  and was applied consistently to all models in our comparison. Lastly, AdamW seems to be the default choice for training on Imagenette, as can be seen from [this documentation page](https://docs.fast.ai/tutorial.imagenette.html) from the Imagenette benchmark creator.
> > > >
> > > >
> > > >
> > > > 3. Our choice to avoid augmentations was methodological: we wanted to first establish the model's baseline capacity to capture higher-order statistics without introducing transformations that could affect the statistical structure of the images in ways that are difficult to quantify (e.g. for the highlighted shear transformation, nonlinear interactions in image content). Regarding specific augmentations, while they can preserve semantic content, their impact on statistical structure is complex and not always beneficial (see for example [Lin et al (2024)](https://jmlr.org/papers/volume25/22-1312/22-1312.pdf)). On a related note, as a further example, the self-supervised learning literature provides numerous references where certain augmentations don't contribute to, or can even harm, representation learning (e.g., [Chen & He, 2021](https://arxiv.org/pdf/2002.05709); [Tian et al., 2020](https://proceedings.neurips.cc/paper/2020/file/4c2e5eaae9152079b9e95845750bb9ab-Paper.pdf)). We believe that the fact that our model achieves competitive performance without augmentations demonstrates its robust feature extraction capabilities. Rather than a limitation, we view this as a strength - achieving strong performance without requiring additional data transformations, while leaving room for potential further improvements through carefully chosen augmentations. Specifically: future work will start from a solid baseline to then understand which data augmentation can work for our models. Lastly, not adding any augmentation makes the perturbation analysis even more relevant, by highlighting to which image statistics each architecture relies.
> > > >
> > > > We hope to have exhaustively addressed your points.

---

> > > > > ### Comment · Reviewer_ELAH · 2024-12-03
> > > > > **Thanks for the reply**
> > > > >
> > > > > I thank the authors for the clarification. I am likely to be off the mark regarding anything about Imagenette, since I was basing my comments on ImageNet. The authors have attempted to include a larger dataset, which directly addresses my initial criticism, so I will slightly raise my score.
> > > > >
> > > > > Regarding the second point, I strongly believe that the authors cannot make any claims about achieving "competitive performance without augmentations", when there are no experiments done with augmentations. The authors started by saying that they omitted augmentations to better understand the mechanisms underlying higher-order statistics, and now are giving bunch of reasons that I strongly believe (in my opinion) were never considered in the beginning such as perturbation analysis. They are not entirely contradicting, but all I see is arguing to make their work look good against the spirit of research. My stance remains the same -- augmentations have not been adequately explored.
> > > > >
> > > > > I wish the authors good luck.

---

> > > > > > ### Author Response · Authors · 2024-12-04
> > > > > > **Final Comment**
> > > > > >
> > > > > > Thank you for your follow-up comments and for acknowledging our efforts to address your initial criticism by including a larger dataset. We appreciate your decision to slightly raise your score in light of these revisions.
> > > > > >
> > > > > > Regarding your second point about data augmentation, we believe there may have been a misunderstanding. We never claimed to achieve performance equivalent to models trained with augmentations. When we say "competitive performance without augmentations," we are specifically referring to our model's performance compared to other architectures (baseline CNN, ResNet-18, VOneNet) under the same conditions - that is, without augmentations.
> > > > > >
> > > > > > Our choice to omit augmentations was straightforward: we wanted to evaluate the baseline capabilities of our architecture compared to standard CNNs without introducing additional factors that could affect the comparison. This allowed us to directly assess the impact of our architectural modifications. Instead, the perturbation analysis you mentioned was conducted to address separate questions raised by another reviewer, and was not related to our initial decision regarding augmentations. This analysis provides valuable insights into how different architectures process image statistics.
> > > > > >
> > > > > > We understand that exploring the impact of data augmentation could be valuable future work. However, our current goal was to demonstrate that our architecture can outperform standard CNNs and biologically-inspired models in a controlled setting where all models are trained under identical conditions.
> > > > > >
> > > > > > We appreciate your engagement with our work and the opportunity to clarify these points.

---

### Official Review · Reviewer_EyLg · 2024-10-30

**Soundness:** 4
**Presentation:** 3
**Contribution:** 4
**Rating:** 6
**Confidence:** 4

**Summary:**

The paper introduces higher-order 2D convolutions with the objective of capturing higher-order dependencies across pixels in natural images. They show convincing results on synthetic, then standard visual datasets.

**Strengths:**

It's very common at ICLR to benchmark any visual model with humoungous datasets of millions of images, and paradoxically I find laudable the choice of focusing on synthetic data and "relatively small datasets to allow for comprehensive testing and analysis". This allows to better explain *why* the model brings something to the community, something often missing in papes just exhibiting accuracy results.

**Weaknesses:**

In figure 6c you use the "Absolute difference between the two RDM" and I would like to question how it brings something qualitative to your analysis... RDMs contains values between 0 and 1 and using a dissimilarity metric based on the distance between probabilities may sound mathematically more justified in comparison to the absolute difference between probability values?

In particular, one main point you mention is "This alignment between model performance and natural image statistics suggests that our approach effectively captures the structure of visual information in the natural world." and I guess it would be more sensible to show how well you capture these statistics instead of using a RDM which focuses on capturing similarities across intermediate representations.


The paper is quite clear and well organized. Minor typos:
Figure 2: indices should start at i=1 not i=0
l 430 "analized"

**Questions:**

The higher order representation is introduced in the input, but nothing limits from using it in other layers: what do you expect to happen if it introduced at other levels of the hierarchy?

You make a point in section 2.1 about the tied-weight issue. First, this could fit in the intro to justify the use of HO representations. An interesting prospect would be to show how you resolved the problem quantitatively in your model. Could you provide quantitative evidence demonstrating how your model resolves the tied-weight issue? In particular, a receptive field captures some higher-order correlations between neighbouring pixels. Would your results still be valid if you would increase kernels' sizes?



As a perspective, a similar architecture could be applied to signal evolving in the temporal domain. Would the higher-order representation be akin to the different moments in generalized coordinates (speed, acceleration, ...) and be something useful for predictive processing ?

---

> ### Author Response · Authors · 2024-11-24
>
> Dear Reviewer EyLg,
>
> Thank you for your thoughtful review and constructive feedback. We appreciate the positive assessment of our work, particularly regarding our methodological choice to focus on testing with synthetic and smaller datasets. Here, we address all the key points you've raised. The main novelties are the addition of a novel perturbative analysis (akin to adversarial attacks) to show how HoCNN captures higher order correlations in natural images as well as a quantitative assessment of the "tied-weight issue". Additionally, in response to the other reviewers, we will also include an analysis on a larger dataset as Imagenette. Finally, we have corrected the noted typos in Figure 2 (indices starting at i=1) and line 430 ("analyzed").
>
> **Better metric for comparison between RDMs**
>
> Thank you for your suggestion regarding the comparison of RDMs. We agree that using absolute differences between probability-like values may not be the most mathematically justified approach. In response, we have revised our analysis using the Hellinger distance, which is particularly suitable for comparing values in the [0,1] range as it satisfies key metric properties (symmetry, triangle inequality) while being bounded. Additionally, we complement this analysis with a log-ratio visualization, which offers an intuitive interpretation of relative changes between models while being scale-invariant. The combination of these two metrics provides a more rigorous mathematical framework for comparing RDMs: the Hellinger distance captures the absolute dissimilarity between the representations, while the log-ratio highlights the relative improvements and helps identify where HoCNN shows meaningful enhancements over the baseline CNN model.
>
> As you suggested, we have also developed a method to quantify how better our HoCNN captures higher-order image statistics (see following paragraph).We believe that combining both approaches - RDM analysis and image statistics quantification - offers a complete analysis and understanding of how HoCNN leads to better image classification performances.
>
> ------
> **Addressing HoCNN's Sensitivity to Higher-Order Image Statistics**
>
> To address how the HoCNN better captures higher-order statistics, we present an alternative analysis that directly quantifies HoCNN's sensitivity to higher-order image statistics through systematic perturbation analysis.
>
> Our methodology employs adversarial attacks (perturbations) on the CIFAR-10 test dataset (10,000 images), using the 50 pretrained models for both the baseline CNN and HoCNN. We generated synthetic textures with varying orders of statistical correlations (1-, 2-, 3-, and 4-point, similar to our synthetic dataset) and added them to the original test images at different intensity levels (ranging from 0.05 to 0.20).
>
> While HoCNN outperforms CNN on clean images (CIFAR-10 test accuracy (%): CNN: 69.76 ± 0.64%; HoCNN: 72.87 ± 0.54%, see manuscript), its performance drops more severely when images are perturbed with higher-order statistical textures (see tables below, in **bold** largest decrease in performance - accuracy (%) - across models).
>
> * CNN Model accuracy (%) for each perturbation type and for different intensities I, in parentheses the decrease in accuracy with respect to the case without perturbation  (CIFAR-10 avg test accuracy = 69.76 %):
>
> | Perturbation | I=0.05 | I=0.09 | I=0.12 | I=0.16 | I=0.20 |
> |-------|---------|---------|---------|---------|---------|
> | γ    | 67.0 (**-4.0%**) | 59.9 (-14.1%) | 50.5 (-27.6%) | 41.4 (-40.7%) | 34.1 (-51.2%) |
> | β-   | 66.5 (-4.7%) | 59.7 (-14.5%) | 50.4 (-27.7%) | 40.2 (-42.3%) | 31.2 (-55.2%) |
> | β\|  | 57.8 (-17.1%) | 41.5 (-40.5%) | 30.8 (-55.8%) | 24.6 (-64.7%) | 21.0 (-69.9%) |
> | β\\  | 55.5 (**-20.5%**) | 31.2 (-55.2%) | 18.0 (-74.2%) | 13.2 (-81.0%) | 11.8 (-83.1%) |
> | β/   | 56.6 (-18.9%) | 32.6 (-53.2%) | 19.0 (-72.8%) | 13.9 (-80.1%) | 12.2 (-82.5%) |
> | θ◢   | 54.2 (-22.2%) | 30.9 (-55.6%) | 18.6 (-73.3%) | 13.8 (-80.2%) | 11.9 (-82.9%) |
> | θ◤   | 54.4 (-22.0%) | 31.2 (-55.3%) | 18.7 (-73.1%) | 13.9 (-80.1%) | 11.9 (-82.9%) |
> | θ◥   | 54.3 (-22.1%) | 31.0 (-55.6%) | 18.8 (-73.1%) | 13.9 (-80.0%) | 12.0 (-82.7%) |
> | θ◣   | 54.2 (-22.3%) | 30.7 (-55.9%) | 18.5 (-73.5%) | 13.7 (-80.3%) | 11.9 (-83.0%) |
> | α    | 56.1 (-19.6%) | 36.0 (-48.4%) | 24.0 (-65.7%) | 18.3 (-73.7%) | 15.8 (-77.3%) |

---

> ### Author Response · Authors · 2024-11-24
>
> * HO-CNN Model (CIFAR-10 avg test accuracy = 72.87 %) :
>
> | Perturbation | I=0.05 | I=0.09 | I=0.12 | I=0.16 | I=0.20 |
> |-------|---------|---------|---------|---------|---------|
> | γ    | 70.2 (-3.6%) | 62.5 (**-14.2%**) | 51.9 (**-28.8%**) | 41.2 (**-43.4%**) | 32.8 (**-55.0%**) |
> | β-   | 69.3 (**-4.9%**) | 61.7 (**-15.3%**) | 52.1 (**-28.5%**) | 41.3 (**-43.3%**) | 31.7 (**-56.5%**) |
> | β\|  | 58.5 (**-19.8%**) | 40.3 (**-44.8%**) | 29.1 (**-60.0%**) | 23.2 (**-68.2%**) | 20.2 (**-72.3%**) |
> | β\\  | 58.5 (-19.7%) | 30.6 (**-58.0%**) | 16.5 (**-77.4%**) | 12.6 (**-82.7%**) | 11.7 (**-84.0%**) |
> | β/   | 58.8 (**-19.3%**) | 30.5 (**-58.2%**) | 16.4 (**-77.5%**) | 12.5 (**-82.8%**) | 11.7 (**-83.9%**) |
> | θ◢   | 56.3 (**-22.7%**) | 29.1 (**-60.1%**) | 16.0 (**-78.0%**) | 12.1 (**-83.4%**) | 10.9 (**-85.0%**) |
> | θ◤   | 56.3 (**-22.8%**) | 29.1 (**-60.0%**) | 16.2 (**-77.8%**) | 12.2 (**-83.2%**) | 11.0 (**-84.9%**) |
> | θ◥   | 56.4 (**-22.6%**) | 29.4 (**-59.6%**) | 16.4 (**-77.5%**) | 12.3 (**-83.1%**) | 11.0 (**-84.9%**) |
> | θ◣   | 56.3 (**-22.8%**) | 29.1 (**-60.1%**) | 16.1 (**-77.9%**) | 12.2 (**-83.3%**) | 11.0 (**-85.0%**) |
> | α    | 58.4 (**-19.8%**) | 35.8 (**-50.9%**) | 22.5 (**-69.2%**) | 17.0 (**-76.6%**) | 14.9 (**-79.6%**) |
>
>
> This greater vulnerability to statistical perturbations increases as we move from lower to higher-order correlations, with the performance gap becoming particularly pronounced at higher perturbation intensities. This stronger degradation in performance provides strong evidence that HoCNN's superior performance on the CIFAR-10 test accuracy stems from its enhanced ability to leverage higher-order statistical features. We are adding this analysis along with a new figure to the paper that thoroughly documents these findings. We sincerely appreciate your constructive feedback that helped us demonstrate more rigorously how HoCNN's improved classification performance is grounded in how it can better capture natural image statistics.
>
> ---
> **Placement of Higher-Order Layers and Effects of Kernel Size**
>
> Thank you for your interesting questions about the placement of higher-order operations in the network hierarchy and the effects of kernel size in capturing higher-order correlations. We conducted extensive experiments to investigate this, testing both alternative placements of the higher-order block and the effect of larger kernel sizes in standard CNNs.
>
> First, regarding the placement of higher-order operations: we tested introducing the higher-order block at the second and third convolutional layers for both CIFAR-10 and CIFAR-100. Our results show that performance slightly degrades as the higher-order block is placed deeper in the architecture: for CIFAR-10 we get 71.17 ± 0.51 % accuracy (%) when introducing the higher-order block at the 2nd convolutional layer and 70.29 ± 0.49 % when moving it to the 3rd convolutional layer; while for CIFAR-100 we get 37.27 ± 0.80 % in the first case and 36.09 ± 0.81 % in the second case. We hypothesize that this happens because earlier layers may already capture and potentially overfit to spurious correlations (related to “tied-weights issue”, see paragraph below), making it harder for later higher-order operations to extract meaningful statistical structure. While standard convolutions theoretically preserve higher-order correlations, the combination with nonlinearities and pooling operations can introduce and amplify spurious correlations that affect the network's ability to learn meaningful representations.
>
> Second, we investigated whether simply increasing kernel sizes in standard CNNs could achieve similar benefits. While larger kernels can theoretically capture higher-order correlations (as evidenced by improved performance of standard CNNs with larger kernels on our synthetic texture dataset), our experiments with 5×5 and 7×7 kernels on both CIFAR-10 and CIFAR-100 show decreased test accuracy compared to our baseline CNN: for CIFAR-10 we get 68.72  ± 0.63 % accuracy (%) for 5x5 kernels and 67.51 ± 0.61 % for 7x7 kernels; while for CIFAR-100 we get 36.04 ± 0.76 % for 5x5 kernels and 34.01 ± 0.72 % for 7x7 kernels. It is worth noting that our choice of 2×2 kernels for the main comparison was deliberate, as it matches the generative process of our synthetic textures, which are based on gliders up to 4 pixels arranged in 2×2 patches. Our results suggest that while larger kernels might have the capacity to capture higher-order interactions, the linear nature of standard convolutions makes it challenging to effectively learn these patterns during optimization. Our results indicate that explicitly modeling higher-order interactions through dedicated operations, as in HoCNN, provides a more effective approach for capturing and leveraging these statistical features.

---

> > ### Author Response · Authors · 2024-11-24
> >
> > These findings support our architectural choice of introducing higher-order operations early in the network, where they can directly process raw image statistics before they are potentially distorted by successive layers of processing. Additionally, this choice is computationally advantageous: in typical convolutional architectures, the number of channels is smaller in earlier layers. Introducing higher-order kernels later in the hierarchy, where the number of channels is larger, would substantially increase the number of parameters and computational complexity of the model.
> >
> > ----
> > **A Quantitative Perspective on the Tied-Weight Issue**
> >
> > Thank you for this insightful suggestion about the tied-weight issue. We agree that this point could be emphasized earlier in the paper and have moved it to the introduction. More importantly, we have conducted a quantitative analysis to demonstrate how HoCNN addresses this issue, which we are adding to the main paper.
> >
> > Our analysis involves the following methodology:
> >
> > 1. We generated a fixed (32 x 32 pixels) input image containing binary textures that exhibit all possible 1-, 2-, 3-, and 4-point correlations for 2×2 patches, corresponding to fixing the x_i terms in Equation (4) of the paper.
> >
> > 2. We initialized the same model architecture 10,000 times and focused on the activations after the convolutional block (Conv - 10, 2x2 kernels - + BatchNorm + ReLU + Max Pooling) or higher-order convolutional block (HoConv - 2, 2x2 kernels - + BatchNorm + ReLU + Pooling), effectively generating many different combinations of the w_i terms.
> >
> > 3. We performed PCA on those activations and measured the number of components needed to explain 95% of the variance
> >
> > The underlying hypothesis is that the tied-weight effect should decrease the number of Principal Components (PCs), while, for networks with larger freedom in the coefficients, the amount of PCs should be high. Our results, using ReLU nonlinearity as in our main architecture, show that, for 95% explained variance :
> >
> > * Baseline CNN requires 0.9% of the components (70 out of 7688)
> > * HoCNN with 2nd order expansion requires 5.3% of the components (102 out of 1922)
> > * HoCNN with 3rd order expansion requires 8.3% of the components (159 out of 1922)
> >
> > Note that CNN has four times more total componentes because it has 4 times more channels. The same analysis done after matching the number of channels gave very similar results.
> >
> > These results quantitatively demonstrate that higher-order convolutions introduce more independent weights, leading to a richer representation space with less weight tying. The increasing number of components needed for variance explanation directly show how our approach addresses the tied-weight issue. We are adding these analyses and corresponding figures to the main paper.
> >
> > Additionally, we conducted the same analysis with different nonlinearities (including Leaky Relu, Gelu, Sigmoid and the more modern Mish), which showed consistent results, further supporting our conclusions. These results on additional nonlinearities will be included in the appendix.
> >
> > **Perspectives on Spatio-temporal correlations**
> >
> > Lastly, we really appreciate your observation about temporal applications: we are particularly happy to expand on this point. The statistical field theory approaches to motion estimation by Potters & Bialek (1994) demonstrated that Bayes-optimal motion estimators naturally evolve into nonlinear operators that adapt to environmental parameters such as contrast and correlation time. Their work shows how higher-order correlations in temporal signals, similar to our spatial higher-order convolutions, emerge as fundamental components of optimal processing. Clark et al. (2015) further expanded this framework, demonstrating how these higher-order terms can be mapped not only to generalized coordinates but also to specific types of motion, including translational movement, approaching and receding motions, and affine transformations. This mapping provides a rich theoretical foundation for understanding how different orders of processing relate to specific types of natural motion.
> >
> > Our architecture could indeed be adapted for temporal signal processing, where the higher-order representations would capture increasingly complex temporal dependencies. This could be particularly valuable for predictive processing tasks, where understanding higher-order temporal statistics (beyond simple linear trends) is crucial for accurate forecasting.
> >
> > To conclude, we are adding all these responses in the revised version of our paper.
> >
> > Best regards

---

> > > ### Comment · Reviewer_EyLg · 2024-11-27
> > > **Acknowledgement of author comments**
> > >
> > > I thank the authors for their very comprehensive response.
> > >
> > > There are of course lots of questions and open issues, many raised in parallel by other reviewers, and I would like to stress on one point: do you have a qualitative assessment of the type of non-linear statistics that you capture in your network?

---

> > > > ### Author Response · Authors · 2024-11-29
> > > >
> > > > Thank you for this follow-up. Could you please clarify what you mean by "non-linear statistics"? If you are referring to higher-order correlations, we think this aspect is addressed in our work through complementary perspectives, following your first set of questions in the previous response:
> > > >
> > > > The quantitative perspective on the tied-weight issue (Fig 1B and subsection 2.1 in the revised paper) explains why the higher-order networks have more expressive power, while the perturbation analysis (Figure 4A and B and Section 4) provides qualitative insights into how this power is utilized. Detailed quantitative results can be found in Tables 3 and 4 in Appendix A.6.
> > > >
> > > > Concisely, perturbations of 2nd, 3rd, 4th order statistics type are degrading HoCNN performances more than with respect to the baseline CNN model, meaning that our network is more sensitive to these kinds of statistics.
> > > >
> > > > We would be grateful for any clarification if you had a different type of non-linear statistics in mind. We also welcome the opportunity to address the “lots of questions and open issues” you see in our work, as this dialogue helps us improve and strengthen our contribution.

---

### Official Review · Reviewer_Rcmj · 2024-11-01

**Soundness:** 2
**Presentation:** 3
**Contribution:** 1
**Rating:** 3
**Confidence:** 4

**Summary:**

In this manuscript, the authors present their new architecture variant HoCNN, which they use as a replacement for the first layer in a simple convolutional neural network architecture. HoCNN adds weights for the higher order terms, i.e. the products of the pixel values, such that the overall function becomes an arbitrary polynomial in the input pixel values for each patch. They find good performance of the network for categorisation of textures from a classical model based on the same correlations HoCNN is build to detect and slight increases in performance for small machine learning datasets (MNIST, FashionMNIST, CIFAR-10, CIFAR-100). Furthermore the authors present some analysis of the distances between stimuli in the network.

**Strengths:**

Extensions of neural networks to more non-linear behaviour are certainly of interest and the authors do find some improvements with their new architecture.

**Weaknesses:**

Overall, I do not think this architecture is promising based on the data the authors present and do not think any of the author’s claims about biological inspiration hold up.

First on the biological inspiration: The type of nonlinearity the authors use here is a general purpose approximation for any nonlinear function and was introduced as such originally. Volterra kernels are an early idea to capture non-linear functions at all, but they are not a good way to capture biologically plausible non-linearities. The receptive fields and non-linear interaction over space in early visual processing are well understood and look nothing like the classical polynomial proposed by the authors. And conversely the authors method allows the approximation of any non-linear function including ones very far from biological plausibility.

In terms of raw performance the authors’ model is not very impressive. They only evaluate on small datasets and with the exception of MNIST the performance of both their model and the CNN they compare to is far from the state of the art. Even for these comparisons HoCNN wins by a few percentage points. This is by no means convincing evidence that their HoCNN really improves performance. For example, according to the FashionMNIST website HOG features and a SVM or a vanilla 2 layer convnet without preprocessing both perform better than the networks presented here (at ~92.5%).

The overall number and complexity of the evaluations is lacking. To establish a new architecture the authors should test different scales of their model, different training environments, etc. and show that their advantage is not just a side effect of a slightly better match in training parameters. Also, it would be important to see comparisons to models that perform reasonably well on the tasks to alleviate doubts like: Is the addition helpful because the convnet is too simple in itself? If so, does the advantage go away if you use a deeper network afterwards?

To make the argument the authors are trying to make (i.e. using this particular image processing frontend based loosely on biology helps ML) they would have to compare to models like VOneNet (https://github.com/dicarlolab/vonenet, https://proceedings.neurips.cc/paper/2020/file/98b17f068d5d9b7668e19fb8ae470841-Paper.pdf ).

**Questions:**

I am still quite uncertain what we can learn from the RSA analyses. What insight is provided by similarities between stimuli are generally lower in HoCNN than in the CNN?

---

> ### Author Response · Authors · 2024-11-23
>
> Dear Reviewer Rcmj,
>
> Thank you for your thoughtful and detailed review of our manuscript. Your insights have been valuable in helping us improve our work, and we are currently incorporating your suggestions into an updated version. Below, we address the key points you've raised. The main novelties are the addition of a test on a more complex dataset (Imagenette) and a novel perturbation analysis to show how HoCNN capture higher order correlations in naturalistic images.
>
> **Biological Inspiration**
>
> Regarding biological inspiration, we take on board your critique of our use of Volterra kernels. While they may be general-purpose approximations, our motivation stems from extensive evidence of non-pointwise processing in biological visual systems, from retinal cells (Lettvin et al., 1959) to cortical neurons (Hubel & Wiesel, 1965), across various species (Fitzgerald & Clark, 2015). Indeed, this kind of nonlinear processing becomes particularly relevant for spatiotemporal stimuli, as also noted by Reviewer EyLg.
>
> There exists a deep connection between different orders of the Volterra approximation and different moments in generalized coordinates (speed, acceleration, etc.). This connection has been elegantly demonstrated in the context of Bayes-optimal motion estimators by Potters & Bialek (1994), who showed how statistical field theory naturally gives rise to nonlinear estimators that adapt to environmental parameters like contrast and correlation time. While these mechanisms seem particularly suited to the spatiotemporal domain, Koenderink & Van Doorn's work on Scale-Space theory demonstrates that similar principles hold for static images as well.
>
> Neuronal systems demonstrate sophisticated computational mechanisms that go beyond simple pointwise nonlinearities, particularly in early visual processing. Our choice of higher-order convolutions represents an **algorithmic implementation** of these biological findings, attempting to capture non-pointwise computations observed in nature (as detailed in *Section 1.2* of the paper). While we acknowledge there could be other, potentially more biologically plausible approaches, our goal was to translate the biological evidence of complex spatial integration and non-pointwise processing into a computational framework. We agree that our model's evaluation should focus on its computational performance while maintaining transparency about the relationship between our implementation and the biological principles that inspired it. We are happy to formulate these points more clearly and have already begun incorporating these clarifications into our revised manuscript
>
> ----
> **Performance Evaluation & Complexity**
>
> Concerning performance evaluation, we want to clarify that achieving state-of-the-art results was not our primary objective. Instead, we focused on conducting fair comparisons between architectures to isolate the impact of higher-order interactions, deliberately avoiding the optimization of multiple factors like augmentations or pre-training protocols that could confound our analysis.
>
> To address your concern about deeper networks and different scales, we have conducted additional experiments with more complex architectures. We implemented Higher-order Convolution in a ResNet-18 architecture (HoResNet-18) and evaluated it on  [Imagenette](https://github.com/fastai/imagenette)  (a subset of ImageNet with 10 classes), achieving 89.30% test accuracy compared with 88.13% for standard ResNet-18. These results demonstrate that the benefits provided by our approach persist with deeper architectures and more challenging datasets.
>
> For completeness here we report architectural details and parameter count:
>
> 1. Standard ResNet-18 architecture totalling 11181642 parameters
>
> * Initial Block:
>     * A convolutional layer (7×7 kernels, 64 channels)
>     * Batch normalization
>     * ReLU activation
>     * Max pooling (3×3)
>
> * Four main stages, each containing 2 residual blocks:
>     * Stage 1: 64 channels (2 blocks)
>     * Stage 2: 128 channels (2 blocks)
>     * Stage 3: 256 channels (2 blocks)
>     * Stage 4: 512 channels (2 blocks)
>
> * Final layers:
>
>     * Global average pooling
>     * Fully connected layer to output classes
>
> 2. Higher-order Resnet-18 (Ho-Resnet-18), totalling 11168382 parameters. It follows a similar structure wrt Resnet-18 with a key modification in **bold**
>
> * Initial Block (similar to ResNet-18):
>     * Convolutional layer (7×7 kernels, **30 channels**)
>     * Batch normalization
>     * ReLU activation
>     * Max pooling (3×3)
>
> * Four main stages, but with a hybrid approach:
>     * **Stage 1: 30 channels with higher-order residual blocks (2 blocks)**
>     * Stage 2: 128 channels with standard residual blocks (2 blocks)
>     * Stage 3: 256 channels with standard residual blocks (2 blocks)
>     * Stage 4: 512 channels with standard residual blocks (2 blocks)
>
> * Final layers (similar to ResNet-18):
>     * Global average pooling
>     * Fully connected layer to output classes

---

> > ### Author Response · Authors · 2024-11-23
> >
> > Additionally, the training setup for our experiments can be summarized as follows:
> >
> > * Data Preprocessing:
> >     * *Training images*: Random resized crop to 224×224, random horizontal flip, normalization with mean and std of [0.5, 0.5, 0.5]
> >     * *Test images*: Resize to 256×256, center crop to 224×224, normalization with mean and std of [0.5, 0.5, 0.5]
> >
> > * Training Configuration:
> >     * Batch size: 64
> >     * Loss function: Cross-entropy
> >     * Optimizer: AdamW with learning rate 0.001 and weight decay 5e-4
> >     * Learning rate scheduling: ReduceLROnPlateau (halves LR after 5 epochs without improvement)
> >     * Early stopping: Implemented with 12 epochs patience
> >
> > ----
> > **VOneNet Comparison**
> >
> > Regarding the VOneNet comparison, we compared our HoResNet-18 model with VOneNet (adapted to ResNet-18 architecture and trained from scratch). Our model achieved better results with fewer parameters (11,168,382 vs. 14,445,322), reaching 89.30% accuracy compared to VOneNet's 88.02%. We will include this comparison in the manuscript.
> >
> > This result, however, could be  expected. The VOneNet authors themselves report lower ImageNet accuracy compared to standard ResNet architectures while achieving better brainscore (i.e., higher explained variance of neural activity recorded from specific brain areas in the visual cortex). This can be verified on the [Brain-Score framework website](www.brain-score.org/vision/) by sorting models based on ImageNet-top1 performance.
> >
> > ----
> > **RSA Analysis Question**
> >
> > Regarding the RSA analyses, the lower similarities between stimuli in HoCNN suggest that our model may be learning more discriminative representations, potentially contributing to its improved performance. We acknowledge that this is an indirect measure of performance improvement and, as already noted by Reviewer EyLg, it does not directly address how well the network captures underlying image statistics. To address this, we conducted additional experiments where we perturbed CIFAR-10 test images with low-intensity textures having specific statistical correlations (similar to our toy dataset but matching CIFAR-10 image dimensions): if a network uses higher-order correlations to learn the classes, these perturbation should hinder its performance. To test for this we generated 10 different test sets, by adding to the normal images a set of small pixel perturbations with a given correlation structure . As expected, when testing both the baseline model and HoCNN (trained on the unperturbed CIFAR-10) across these artificial test sets, we consistently found that HoCNN reports larger test errors across different perturbation values (with more pronounced errors for 2nd and 3rd order statistics perturbations). This directly confirms that its image representations capture higher-order image statistics more effectively than a standard CNN.
> >
> > We are currently including this analysis in the revised manuscript to provide deeper insights into the learned representations.
> >
> > Best regards,

---

> > > ### Comment · Reviewer_Rcmj · 2024-11-26
> > > **Acknowledgement of author comments**
> > >
> > > I wanted to acknowledge the authors responses and provide comment on them: I think the authors are moving in the right direction by providing evaluations on more complex datasets & models. However, the differences to other models remain tiny, it remains unclear what parts should actually be replaced as we are now moving deeper into the networks for some reason, and the evaluations are still done on a relatively obscure smaller subset of the data instead of the real ImageNet. Thus, I remain unconvinced that using the higher order kernels is a good idea.
> > >
> > > Also, I actually know some of the classical literature on these types of Kernels and nonlinear systems to estimate properties like motion. I agree with the authors that there is fairly good evidence that there are nonlinear interactions over space already in the retina and certainly in cortex as well. However, these non-linear interactions are not easily representable as Volterra kernels. Even the paper cited by the authors mention adaptations to SNR, contrast, brightness, etc. which all require complex adaptations of the weights if the transformations are expressed as Volterra Kernels. If the authors had included some kind of contrast / brightness / SNR / whatever dependent rescaling of interactions, this would be an entirely different conversation. For simply using higher inner products, I really don't see a strong biological motivation.
> > >
> > > So on neither front, I am convinced of this approach and will not change my scores. Nonetheless, I wish the authors good luck with this endeavour. It would be great to have some more biologically plausible nonlinear interactions, but this is not it yet.

---

> > > > ### Author Response · Authors · 2024-12-02
> > > >
> > > > Thank you for your positive feedback on our work. In the following, we would like to respectfully address the additional points you have raised:
> > > >
> > > > Regarding the "tiny" differences: We believe this characterization warrants further discussion. In the context of [CIFAR-10](https://paperswithcode.com/sota/image-classification-on-cifar-10) and, [CIFAR-100](https://paperswithcode.com/sota/image-classification-on-cifar-100) benchmarks, improvements of comparable or even smaller magnitude are considered significant advances. Could you please clarify your criteria for what constitutes a meaningful improvement?
> > > > About the placement of higher-order layers: following the comment of Reviewer EyLg, we have actually investigated this systematically, and included the analysis into the manuscript (Appendix A.2.1.). Our experiments show a clear pattern: performance decreases when higher-order blocks are placed deeper in the architecture. Specifically:
> > > > * CIFAR-10:
> > > >     * 1st convolutional layer (baseline HoCNN): 72.87 ± 0.54% accuracy
> > > >     * 2nd convolutional layer: 71.17 ± 0.51% accuracy
> > > >     * 3rd convolutional layer: 70.29 ± 0.49% accuracy
> > > > * CIFAR-100:
> > > >     * 1st convolutional layer (baseline HoCNN): 40.42 ± 0.84% accuracy
> > > >     * 2nd convolutional layer: 37.27 ± 0.80% accuracy
> > > >     * 3rd convolutional layer: 36.09 ± 0.81% accuracy
> > > >
> > > > We hypothesize this occurs because earlier layers may capture and potentially overfit to spurious correlations, making it harder for later higher-order operations to extract meaningful statistical structure.
> > > > Regarding your characterization of Imagenette as "obscure": We respectfully disagree with this assessment. Imagenette has been widely used as a benchmark in [numerous publications](https://paperswithcode.com/dataset/imagenette), including several in major ML conferences. Could you please elaborate on why you consider it obscure?
> > > > To focus now on the biological plausibility, we agree that achieving truly biologically plausible models requires multiple steps and considerations. We view our current work as a meaningful first step towards biological inspiration for a different architecture.
> > > >
> > > > It's worth noting that standard CNNs, while not fully biologically plausible, have long been considered biologically inspired and recognized as the leading deep learning architecture for modeling primate vision (e.g., Cichy et al., 2019; Kubilius et al., 2019). As demonstrated in our paper, a standard CNN can be interpreted as the first-order expansion of the Volterra kernel (see Equation 2). Our approach builds upon this foundation, inheriting CNNs' established inductive biases while extending their capability to model nonlinear phenomena. The concern about lacking adaptations to contrast, brightness, SNR, and other dynamic rescaling interactions applies equally to standard CNNs. Therefore, we believe our model represents a reasonable step forward in the biological inspiration direction, even though we acknowledge there is still significant room for incorporating more sophisticated biological mechanisms in future work.

---

### Official Review · Reviewer_WTqU · 2024-11-04

**Soundness:** 2
**Presentation:** 3
**Contribution:** 2
**Rating:** 3
**Confidence:** 4

**Summary:**

This paper proposes a novel approach to enhance convolutional neural networks (CNNs) by introducing higher-order convolutions (HoConv) inspired by biological visual systems. The method aims to capture complex pixel relationships explicitly by incorporating second-order and higher-order interactions in the convolutional process. The authors validate the effectiveness of HoConv on synthetic datasets and standard benchmarks like CIFAR-10 and CIFAR-100, demonstrating some performance improvements. Representational similarity analysis (RSA) is employed to investigate the representational differences between HoConv and standard convolutional layers.

**Strengths:**

1.	The introduction of higher-order convolutions is a novel idea, grounded in biological visual processing, which could inspire future work in feature representation.
2.	The theoretical foundation using Volterra series adds rigor to the methodology.
3.	The use of representational similarity analysis (RSA) to explore differences in feature representation provides additional insight into the method’s internal workings.

**Weaknesses:**

1.	The experimental setup is overly simplistic, relying heavily on toy datasets and limited architectures, making it difficult to assess the method’s effectiveness in real-world scenarios.
2.	There is no exploration of how this method could scale to deeper networks (e.g., ResNet) or more complex tasks (e.g., ImageNet), raising concerns about its applicability to complex tasks.

**Questions:**

1.	Could the authors provide a more extensive evaluation of HoConv on deeper networks or larger datasets to demonstrate its scalability and generalizability?
2.	How does the explicit modeling of higher-order interactions compare with the traditional method of adding more layers in terms of computational cost and performance improvement?
3.	Can the authors clarify if and how the architectural differences were controlled in the experiments, especially in Table 1, to ensure a fair comparison?

---

> ### Author Response · Authors · 2024-11-23
>
> Dear Reviewer WTqU,
>
> We sincerely thank you for your detailed review and constructive feedback. Below we address all your comments and questions, while we are currently updating the manuscript to reflect our revisions. Particularly relevant to your previous comments, we now consider the case of a complex dataset (Imagenette) showcasing how our HoCNN again outperforms standard CNN architectures. We hope that these additional results satisfy your previous criticisms and doubts.
>
> **Scalability and real-world applicability (Weaknesses 1 & 2; Question 1)**
>
> We have conducted additional experiments with deeper architectures and more challenging datasets. Specifically, we implemented Higher-order Convolution in a ResNet-18 architecture (HoResNet-18) and evaluated it on [Imagenette](https://github.com/fastai/imagenette) (a subsampled dataset from ImageNet, containing 10 classes), achieving 89.30% test accuracy compared with 88.13% for standard ResNet-18.
>
> These results demonstrate that our approach successfully scales to deeper architectures and more complex datasets, addressing (at least in part) the concern about real-world applicability. Clearly, we cannot test our approach for all possible architectures/datasets currently available, but we hope that our attempt with the specifications above provides a satisfactory indication that our approach does scale effectively. The performance improvement of over 1 percentage point on a challenging dataset further validates the effectiveness of our approach beyond simple toy examples.
>
> We are now adding the analysis of this complex dataset in the revised paper.
>
> For completeness here we report architectural details and parameter count:
>
> 1. Standard ResNet-18 architecture totalling 11181642 parameters
>
> * Initial Block:
>     * A convolutional layer (7×7 kernels, 64 channels)
>     * Batch normalization
>     * ReLU activation
>     * Max pooling (3×3)
>
> * Four main stages, each containing 2 residual blocks:
>     * Stage 1: 64 channels (2 blocks)
>     * Stage 2: 128 channels (2 blocks)
>     * Stage 3: 256 channels (2 blocks)
>     * Stage 4: 512 channels (2 blocks)
>
> * Final layers:
>
>     * Global average pooling
>     * Fully connected layer to output classes
>
> 2. Higher-order Resnet-18 (Ho-Resnet-18), totalling 11168382 parameters. It follows a similar structure wrt Resnet-18 with a key modification in **bold**
>
> * Initial Block (similar to ResNet-18):
>     * Convolutional layer (7×7 kernels, **30 channels**)
>     * Batch normalization
>     * ReLU activation
>     * Max pooling (3×3)
>
> * Four main stages, but with a hybrid approach:
>     * **Stage 1: 30 channels with higher-order residual blocks (2 blocks)**
>     * Stage 2: 128 channels with standard residual blocks (2 blocks)
>     * Stage 3: 256 channels with standard residual blocks (2 blocks)
>     * Stage 4: 512 channels with standard residual blocks (2 blocks)
>
> * Final layers (similar to ResNet-18):
>     * Global average pooling
>     * Fully connected layer to output classes
>
> Additionally, the training setup for our experiments can be summarized as follows:
>
> * Data Preprocessing:
>     * *Training images*: Random resized crop to 224×224, random horizontal flip, normalization with mean and std of [0.5, 0.5, 0.5]
>     * *Test images*: Resize to 256×256, center crop to 224×224, normalization with mean and std of [0.5, 0.5, 0.5]
>
> * Training Configuration:
>     * Batch size: 64
>     * Loss function: Cross-entropy
>     * Optimizer: AdamW with learning rate 0.001 and weight decay 5e-4
>     * Learning rate scheduling: ReduceLROnPlateau (halves LR after 5 epochs without improvement)
>     * Early stopping: Implemented with 12 epochs patience

---

> ### Author Response · Authors · 2024-11-23
>
> ---
> **Computational costs versus traditional methods (Question 2.)**
>
> We compared our HoCNN, which uses Volterra 3x3 kernels only in its first layer (8 channels, combining first, second, and third-order interactions + batch norm + relu nonlinearity and max pooling) followed by two standard convolutional blocks (16 and 32 channels ; + batch norm + ReLU + max pooling) and two fully connected layer interleaved with dropout (p = 0.25), against a deeper CNN with eight convolutional blocks (with 3x3 kernels as well) with increasing channel dimensions (8→8 → 16 → 16 →32 → 32 →64→ 64; + batch norm + ReLU + max pooling) and two fully connected layers interleaved with dropout (p = 0.25). Despite the deeper network having more parameters (108,130 vs 80,262,  that is 25% more) and computational complexity than HoCNN's higher-order operations, our model still achieved superior performance (72.35% vs 71.20% accuracy). Deeper or wider CNNs will eventually reach the HoCNN performance, although at the cost of higher complexity and risk of overfitting. Overall, this suggests that explicit modeling of higher-order interactions in early layers can be more effective than simply increasing network depth or width.
>
> We also provide a detailed analysis of computational complexity: while a standard convolutional layer requires O(Cout × Cin × K² × H × W) FLOPs, second-order kernels scale as O(Cout × Cin × K⁴ × H × W), and third-order as O(Cout × Cin × K⁶ × H × W). Therefore HoCNN layers scale more rapidly with the kernel size K. However, while the computational complexity represents a current limitation of our approach, we envision future improvements through low-rank approximations, similar to successful approaches used in transformer architectures. This would maintain the benefits of higher-order interactions while reducing computational complexity.
>
> ----
> **Architectural differences (Question 3.)**
>
> We apologize if our experimental setup description was not sufficiently clear. To ensure fair comparison, both architectures follow the same basic structure and have a similar number of parameters. They both have two convolutional layers (first layer: 10, 2×2 kernels, second layer: 2, 8×2 kernels) followed by one fully connected layer. The HoCNN variant maintains this structure but employs only 2 kernels in the first layer (instead of 10), expanded to include higher-order terms. This results in a similar parameter count while maintaining architectural similarity. This design choice allows us to directly compare the impact of higher-order interactions while controlling for other architectural factors.
>
> Lastly, we report data preprocessing and training configuration details for the two networks:
>
> * Single-channel normalization with mean 0.5 and std 0.5 (z-score normalization)
> * No data augmentation applied
> * Batch size: 64
> * Loss function: Cross-entropy
> * Optimizer: AdamW with learning rate 1e-3 and minimal weight decay (1e-5)
> * Learning rate scheduling: ReduceLROnPlateau (reduces LR by 80% after 5 epochs without improvement)
> * Early stopping: Implemented with 12 epochs patience
>
> We clarify these details into both the paper and the Appendix.
>
> Best regards,

---

> > ### Author Response · Authors · 2024-12-02
> > **Follow-Up on Previous Response**
> >
> > Dear Reviewer WTqU,
> >
> > We wanted to follow up on our previous response to your comments. We have updated our submission to address the points you raised, incorporating the additional experiments, analyses, and clarifications discussed in our reply.
> >
> > We hope these revisions adequately address your concerns and strengthen our manuscript. If you have any further questions or feedback, we would be more than happy to discuss them with you.
> >
> > Thank you for your time and consideration.

---

### Meta-Review · Area_Chair_CTA1 · 2024-12-22

**Metareview:**

This paper introduces higher order convolutions in CNNs, citing the importance of higher-order statistics from neuroscience literature. The authors report favorable results in MNIST, F-MNIST, CIFAR-10 and CIFAR100. The method is motivated by the complex nonlinear processing in biological vision systems, and aims to capture complex pixel relationships explicitly by incorporating second-order and higher-order interactions in the convolutional process.  The authors validate the effectiveness of HoConv on synthetic datasets and standard benchmarks like CIFAR-10 and CIFAR-100, demonstrating some performance improvements over standard convolutional networks.

Strengths: Extensions of neural networks to more non-linear behaviour are certainly of interest. The method is rigorously introduced and analyzed using the volterra series. The paper shows promising results on simple datasets. The paper is quite clear and well organized.

Weaknesses: The experiments lack exploration of more realistic datasets (largest tested dataset is ImageNette which is just 1% of ImageNet) or more complex architectures (most complex model is a ResNet-18). Comparisons also exclude other standard techniques such as data augmentations and other methods for capturing non-linear dependencies such as self-attention blocks or the cited related work on higher order convolutions (e.g. Zoumpourlis et al. (2017)). This makes it difficult to judge if the advantages shown in the paper will transfer to a more realistic setup, or if they are made obsolete by other techniques that are already standard.

Overall I find the experimental evidence presented insufficient and recommend rejection.

**Additional Comments On Reviewer Discussion:**

The authors addressed many of the criticisms of the reviewers by clarifications and additional experiments. These experiments definitely went in the correct direction by adding experiments on a larger dataset (ImageNette) and a more modern architecture (ResNet-18). However, I agree with the reviewers, that these do still not go far enough to make a convincing case for HoConv.

---

### Decision · Program_Chairs · 2025-01-22

Reject